# Subsystem Non-Invertible Symmetry Operators and Defects

Weiguang Cao[1,2], Linhao Li[2,3], Masahito Yamazaki[1,4], and Yunqin Zheng[1,3]

[1]   Kavli Institute for the Physics and Mathematics of the Universe,
University of Tokyo, Kashiwa, Chiba 277-8583, Japan

[2]   Department of Physics, Graduate School of Science,
University of Tokyo, Tokyo 113-0033, Japan

[3]   Institute for Solid State Physics,
University of Tokyo, Kashiwa, Chiba 277-8581, Japan

[4]   Trans-Scale Quantum Science Institute,
University of Tokyo, Tokyo 113-0033, Japan

We explore non-invertible symmetries in two-dimensional lattice models with subsystem $\mathbb{Z}_2$ symmetry. We introduce a subsystem $\mathbb{Z}_2$-gauging procedure, called the subsystem Kramers-Wannier transformation, which generalizes the ordinary Kramers-Wannier transformation. The corresponding duality operators and defects are constructed by gaugings on the whole or half of the Hilbert space. By gauging twice, we derive fusion rules of duality operators and defects, which enriches ordinary Ising fusion rules with subsystem features. Subsystem Kramers-Wannier duality defects are mobile in both spatial directions, unlike the defects of invertible subsystem symmetries. We finally comment on the anomaly of the subsystem Kramers-Wannier duality symmetry, and discuss its subtleties.

# 1 Introduction

## 1.1 Subsystem non-invertible symmetry

Symmetry is one of the most important notions in physics. In recent years, the concept of global symmetry has been generalized in various directions [1]. See also the snowmass white paper [2] for a recent overview and more references. The three most notable properties of symmetry operators/defects are their

1. codimensions,

2. invertibility,

3. topologicalness/mobility.

We can organize generalized symmetries according to these properties (Tab. 1). An ordinary global symmetry corresponds to codimension-one topological operators with group-like and invertible fusion rules. The remaining symmetries differ in one or multiple properties. For instance,

| Symmetry | codimension | invertibility | topologicalness |
|---|---|---|---|
| ordinary symmetry | $= 1$ | ✓ | ✓ |
| higher-form/group sym [1, 3–45] | $\geq 1$ | ✓ | ✓ |
| non-invertible/categorical sym [50–74] | $= 1$ | ✗ | ✓ |
| higher categorical sym [101–140] | $\geq 1$ | ✗ | ✓ |
| subsystem sym [75–91] | $= 1$ | ✓ | ✗ |
| higher subsystem sym [141–152] | $\geq 1$ | ✓ | ✗ |
| **subsystem non-invertible sym** | $= 1$ | ✗ | ✗ |
| higher subsystem non-invertible sym | $\geq 1$ | ✗ | ✗ |

Table 1: Organization of global symmetries in terms of operator/defect's codimension, invertibility and topologicalness. The goal of this paper is to explore subsystem non-invertible symmetry, highlighted in bold font.

higher-form symmetries relax the condition of codimension being one [1, 3–45];[1] non-invertible symmetries in $(1+1)$d relax the invertibility [50–74]; subsystem symmetries in $(2+1)$d relax the topologicalness or the mobility of the symmetry generators/defects[2] [75–91].[3] Here and in Tab. 1 we only present a highly incomplete list of references.[4]

One gets further generalized symmetries by modifying more than one property simultaneously. For instance, by allowing the defect to have codimension higher than one and also be non-invertible, we get higher categorical symmetries [101–140]; by gauging the 0-form subsystem symmetries in $(3+1)$d, we get a fracton order with one-form subsystem symmetries [141–152].

The goal of this paper is to study yet another type of generalized symmetry by lifting both the invertibility as well as the topologicalness: the corresponding symmetry is a *subsystem non-invertible symmetry*. We will not study a generic subsystem symmetry, and instead focus on one

---

[1]Before [1], various earlier works had already hinted the existence of higher form symmetries and the dynamical consequences thereof [46–49].

[2]Let us clarify the definition of the codimension of an operator/defect when it is not fully mobile. The $(2+1)$d subsystem-symmetry defects are line operators, thus one would naively conclude that it is of codimension two in spacetime, and is higher-form as well. Here it is more natural to define the codimension with respect to the subspace where the operator/defect is mobile rather than the entire spacetime. For instance, consider the subsystem $\mathbb{Z}_2$ symmetry where the defect is along the $x$ direction. It is only mobile along the $t$ direction, while is not mobile along the $y$ direction. So the mobile subspace is the $t - x$ plane, and the defect with respect to the $t - x$ plane is of codimension one.

[3]There are also earlier works on subsystem symmetry from algebraic duality approach [92–96].

[4]Another line of development of unconventional symmetry, motivated by commutant algebra in the study of quantum many body scar, was explored in [97–100]. It would be interesting to see how such unconventional symmetry fits in various entries of Tab. 1.

typical example— the *subsystem Kramers-Wannier (KW) duality symmetry* associated with the gauging of a subsystem $\mathbb{Z}_2$ symmetry in $(2+1)$d. Here, the subsystem KW duality symmetry has *co-dimension 1* non-invertible symmetry operator and defect, which is different from the co-dimension 2 invertible subsystem $\mathbb{Z}_2$ symmetry operators and defects. Furthermore, the non-invertible fusion rule will mix operators (defects) of different co-dimensions.

Our construction naturally generalizes the well-known KW duality symmetry in $(1+1)$d, which gauges an ordinary $\mathbb{Z}_2$ symmetry. To the authors' knowledge, this is the first study of a symmetry where both the invertibility and topologicalness are lifted.

## 1.2 Symmetry operators and defects

For a given symmetry we can consider symmetry operators (along the spatial direction) and symmetry defects (one of whose directions is along the time) [5]. When the symmetry is topological, the two can be deformed to each other and hence are equivalent. This naturally happens in relativistic systems, where Lorentz symmetry ensures that time and space are on equal footing.

However, when the topologicalness is sacrificed as in the case of the subsystem symmetry, it is important to treat the symmetry operator and symmetry defect separately. The latter was highlighted recently [87, 91] as a generator of the time-like symmetry.

In the literature, most of the discussions of non-invertible symmetries are either in continuum field theories or in the statistical models with infinite system sizes [52]. The authors of [154] systematically discussed the properties of the KW-duality operator acting on a finite-size Hilbert space of a quantum system, and carefully discussed the effects of boundary conditions for closed spin chains. However, it is also important to study symmetry defects to fully uncover the properties of symmetry. In particular, it is important to turn on both symmetry operators and defects when determining the anomaly of the symmetry. Anomaly usually manifests itself as a "fractional" symmetry charge of the ground state in the presence of symmetry defects (i.e. twisted boundary conditions) [155, 156].

In this work, we will study both symmetry operators and defects on the lattice. Our discussion of the symmetry operators directly generalizes our previous work on $(1+1)$d KW duality operators [154]. The discussion of symmetry defects is relatively new, and is consistent with earlier results for the $(1+1)$d KW duality defect in the Ising model on the lattice [157].

One peculiar feature we encounter is the mobility of the subsystem KW duality defect. Because of the subsystem nature, one might naively expect that the subsystem KW duality defect along, say, the $t-x$ [6] direction is not mobile along the $y$ direction. However, we find that the

---

[5]We mainly use the terminology "operators" for maps from one Hilbert space (defined on sites) to another Hilbert space (defined on links or plaquettes) and "defects" for interfaces between two theories. One can further redefine the link/plaquette Hilbert spaces to be supported on sites, which we briefly discuss in Sec. 2.4 and 3.4. This redefinition makes the operators to act within on one Hilbert space, and the defects between a single theory. This is consistent with the recent discussion [153].

[6]We use $t$ for both time and twist boundary condition in the main text which should not be confused. In some places, we also use $\tau$ for time, such as in the index of gauge fields and gauge invariant holonomy variables.

subsystem KW duality defect is mobile under translation in the entire $x - y$ plane. Namely, two KW duality defects defined on $t - \ell_1(x, y)$ and $t - \ell_2(x, y)$, where $\ell_i(x, y)$ specifies a pair of mutually deformable curves within the $x - y$ plane, can be deformed to each other.[7]

## 1.3   Anomalies of subsystem (non)invertible symmetry

As with other types of symmetries, the subsystem (non)invertible symmetry may also suffer from anomalies. What does the anomaly of subsystem symmetry mean? We list several criteria, not all of which are equivalent:

1. Incompatible with a gapped phase with one ground state including trivially gapped phase and subsystem symmetry protected topological (SSPT) phases [158–160].

2. Incompatible with a gapped phase constructed from stacking lower dimensional (subsystem) topological quantum field theories (TQFTs).

3. There is a non-trivial anomaly inflow from higher dimensions [161].

In particular, point 2 is more general than point 1, because the concept of $G$ subsystem symmetric Renormalization Group flow is believed to admit stacking layers of lower-dimensional $G$ symmetric TQFTs [162, 163]. Thus it is natural to define the "trivial" phase by modding out the lower dimensional symmetric gapped phases, including those whose ground state degeneracy is higher than one. However, point 2 and point 1 are degenerate in the current case where we only consider a $\mathbb{Z}_2$ subsystem symmetry in $(2 + 1)$d. This is because a non-trivial $\mathbb{Z}_2$-symmetric TQFT in $(1 + 1)$d must spontaneously break the $\mathbb{Z}_2$ symmetry. Thus in the following, we will adopt point 1 as a criterion for the subsystem KW duality symmetry to be anomalous.

The anomaly of the KW duality symmetry in $(1+1)$d has been discussed in [55,56,59,106]. In this paper, we follow the approach in [106] and discuss the anomaly of the subsystem KW duality symmetry in $(2 + 1)$d. The idea is to check whether an SSPT phase is invariant under a gauging of the subsystem $\mathbb{Z}_2$ symmetry; if not, we conclude that the subsystem KW duality symmetry is anomalous.

## 1.4   Structure of the paper

The paper is organized as follows. In Sec. 2, we review the construction of ordinary KW duality operators and defects in $(1 + 1)$d spin chains via gauging ordinary $\mathbb{Z}_2$ symmetries. We present a comprehensive analysis of the ordinary KW duality symmetry, including the derivation of non-invertible fusion rules, the demonstration of the mobility of duality defects, and the proof of the anomalous nature of the KW duality symmetry. In Sec. 3, we extend this formulation to $(2 + 1)$d lattice models, which incorporate subsystem KW duality operators and defects. We conduct a similar investigation of the non-invertible fusion rules, the mobility of duality defects, and the

---

[7]Note that one defect is deformable to another defect, but not to the symmetry operator.

anomaly of subsystem KW duality symmetry. Finally, in Sec. 4, we summarize our findings and suggest potential future directions for this work.

# 2   Kramers-Wannier duality operators and defects in $(1+1)$d spin chains

In this section we briefly review the non-invertible duality symmetry given by the KW transformation with respect to a non-anomalous $\mathbb{Z}_2$ symmetry in $(1+1)$d spin chains. The discussion here serves as a preparation for the subsystem generalization of the non-invertible symmetry in $(2+1)$d, to be discussed in Sec. 3. Most of the materials in this section are scattered in the literature and are not new. The goal is to gather them in one place, and also present them in a way that directly generalizes to the subsystem symmetry case.

## 2.1   Ordinary $\mathbb{Z}_2$ symmetry and twist operators

Consider a $\mathbb{Z}_2$-symmetric theory on either a closed chain with $L$ sites or an infinite chain. On each site $i$ there is a spin-$\frac{1}{2}$ variable $s_i \in \{0, 1\}$. The $\mathbb{Z}_2$ operator is associated with a topological operator with the $\mathbb{Z}_2$ fusion rule. One can either place such a topological operator along the space or along the time direction. In the former case, the topological operator is a symmetry generator acting on the entire Hilbert space,

$$U := \prod_{i \in \mathbb{Z}} \sigma_i^x \,, \tag{2.1}$$

which flips the spin on each site, i.e. $s_i \to 1 - s_i$. The eigenvalue of $U$ is $(-1)^u$ with $u = 0, 1$. The $\mathbb{Z}_2$ symmetry generator satisfies two obvious properties:

1. $\mathbb{Z}_2$ fusion rule: $U \times U = 1$.

2. Topological along the time direction, since it commutes with the Hamiltonian.

In the latter case, the topological operator is a symmetry defect that modifies the Hilbert space. For a quantum field theory in the continuum with Lorentz symmetry, the space and time are on equal footing, and a $\mathbb{Z}_2$ operator and a $\mathbb{Z}_2$ defect are used interchangeably. However, it is useful to distinguish the two, since our discussion is on the lattice, and since we would like to generalize the discussion here to subsystem symmetries.

Let us give further details when $\mathbb{Z}_2$ operators are along the time direction, i.e. the $\mathbb{Z}_2$ defects. It is useful to first discuss $\mathbb{Z}_2$ defects on an infinite chain, and then on a closed chain. Suppose the defect is localized at the origin. In the Hamiltonian formalism, such a defect can be created by acting the original Hilbert space by a *twist operator*, i.e. the $\mathbb{Z}_2$ symmetry generator terminated at

the origin. See Fig. 1 for a schematic explanation. Concretely, the twist operator is[8]

$$U_0^t := \prod_{i \leq 0} \sigma_i^x . \tag{2.2}$$

It creates a $\mathbb{Z}_2$ defect between site $i = 0$ and $i = 1$. Conjugating $U_0^t$ on a $\mathbb{Z}_2$ symmetric Hamiltonian creates a new Hamiltonian $H_{\text{tw},0} := (U_0^t)^\dagger H U_0^t$. For example, start with the $\mathbb{Z}_2$ symmetric Ising model without transverse field, $H = -\sum_{i \in \mathbb{Z}} \sigma_i^z \sigma_{i+1}^z$, then $H_{\text{tw},0} = -\sum_{i \neq 0} \sigma_i^z \sigma_{i+1}^z + \sigma_0^z \sigma_1^z$, and the ground state of the latter supports a $\mathbb{Z}_2$ domain wall. Like the $\mathbb{Z}_2$ symmetry generator, the $\mathbb{Z}_2$ defect also satisfies the same two properties:

1. $\mathbb{Z}_2$ fusion rule. The fusion of $\mathbb{Z}_2$ defects follows from the fusion of two twist operators, $U_0^t \times U_0^t = 1$.

2. Topological along the space direction. Given two defects at $i = 0$ and $i = 1$, generated by $U_0^t$ and $U_1^t$ respectively. They give rise to the twisted Hamiltonians $H_{\text{tw},0}$ and $H_{\text{tw},1}$. Topologicalness of the $\mathbb{Z}_2$ defects means the two Hamiltonians are unitary equivalent, i.e. there is a local unitary transformation $W$ such that $H_{\text{tw},1} = W^\dagger H_{\text{tw},0} W$, where $W$ only acts on the Hilbert space around the origin. This follows if one can find $W$ such that

$$U_1^t = U_0^t W . \tag{2.3}$$

   Indeed, $W = \sigma_1^x$.

We also comment on the $\mathbb{Z}_2$ defect on a closed chain. As explained in Fig. 1, to create a symmetry defect on a ring, we first uplift the circle to an open spin chain with periodic boundary condition $|s_{i+L}\rangle = |s_i\rangle$, and act with the twist operator on the intervals $\cup_{k \in \mathbb{Z}}(2kL, (2k+1)L]$. This changes the periodic boundary condition to the twisted boundary condition $|s_{i+L}\rangle = |s_i + 1\rangle$. Hence on a closed chain, we can use $t = 0, 1$ to label the periodic boundary condition and twisted boundary condition, corresponding to the absence or presence of the $\mathbb{Z}_2$ defect, i.e. $|s_{i+L}\rangle = |s_i + t\rangle$.

## 2.2 KW duality operators

From now on, we assume that the $(1+1)$d system is invariant under the gauging of the $\mathbb{Z}_2$ symmetry. This means that the $\mathbb{Z}_2$ gauging is a symmetry of the system. Since any symmetry is equipped with a topological operator, as we discussed for the $\mathbb{Z}_2$ symmetry in Sec. 2.1, we can put the operator either along the space (i.e. KW duality operator) or along the time (i.e. KW duality defect). In this subsection, we will discuss the KW duality operator.

The KW duality operator acts on the entire Hilbert space via gauging the $\mathbb{Z}_2$ symmetry. We will assume closed boundary conditions throughout this section since the change of (twisted)

---

[8]The superscript $t$ either stands for "time" since the defect is along the time direction, or "twist" since it is a twist operator creating the defect. The subscript 0 labels the endpoint of the twist operator or the locus of the defect.

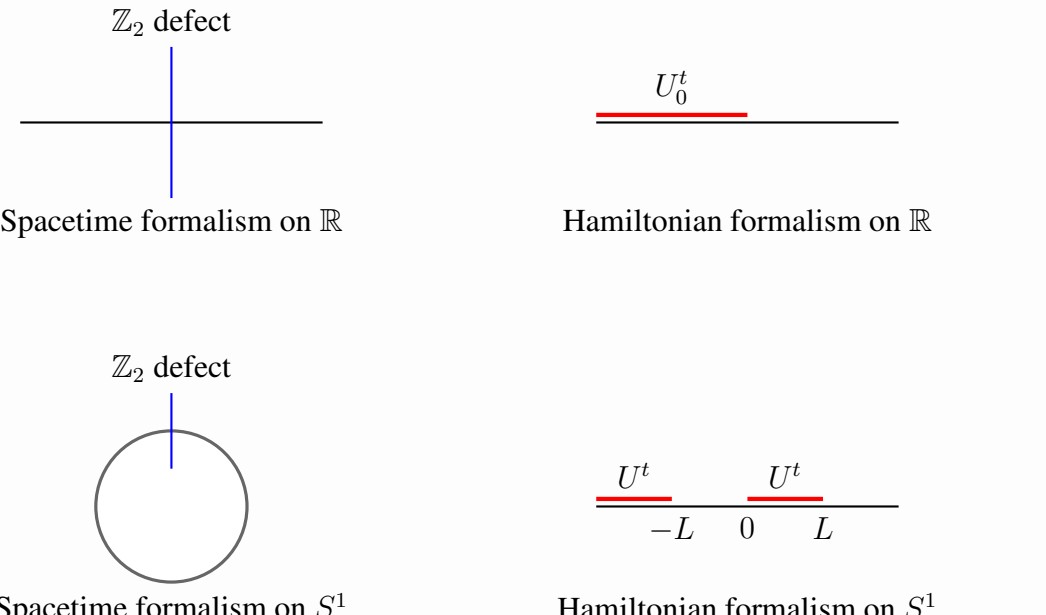

Figure 1: $\mathbb{Z}_2$ symmetry defects (blue) and $\mathbb{Z}_2$ twist operators (red) on an infinite chain and on a ring. The black line represents the space, and the orthogonal direction represents time. In the first row, acting a twist operator on the left half chain creates a $\mathbb{Z}_2$ defect along the time direction at the origin. In the second row, in order to use the twist operator to create a symmetry defect, one needs to lift $S^1$ up to $\mathbb{R}$ and act the twist operator on the intervals $\cup_{k\in\mathbb{Z}}[2kL,(2k+1)L]$.

boundary conditions is a key feature under gauging. The KW duality operator was widely known to map Pauli operators as $\sigma_i^x \to \sigma_i^z \sigma_{i+1}^z$ and $\sigma_i^z \to \prod_{j \leq i} \sigma_i^x$, which is well defined on an infinitely long chain. Its action on the Hilbert space of an infinitely long chain has also been discussed in [52]. See also [164–166] for the of Matrix-Product-State (MPS) realization. However, on a closed chain, one has to specify how the boundary conditions transform under gauging [167–170]. In this subsection, we mainly follow the discussion in [154], where the fusion rule, exchange of symmetry and twist sectors, and mapping of operators are all discussed on closed chains. We only briefly summarize the results, and refer the interested reader to [154] for further details.

Under KW, the spins $\{s_i\}$ on the original lattice are mapped to dual spins $\{\widehat{s}_{i-\frac{1}{2}}\}$ on the link. We use the $(\widehat{u}, \widehat{t})$ to label the symmetry-twist sectors of the dual lattice. Here, $(-1)^{\widehat{u}}$ is the eigenvalue of the dual symmetry $\widehat{U} := \prod_{i=1}^{L} \widehat{\sigma}_{i-\frac{1}{2}}^x$, and $\widehat{t}$ labels the boundary condition $|\widehat{s}_{i-\frac{1}{2}+L}\rangle = |\widehat{s}_{i-\frac{1}{2}} + \widehat{t}\rangle$.

**Definition of KW duality operator:** The KW transformation is realized by an operator $\mathcal{N}$ acting on the Hilbert space. It is sufficient to specify how it acts on the basis states [154]

$$\mathcal{N} |\{s_i\}\rangle = \frac{1}{2^{\frac{L}{2}}} \sum_{\{\widehat{s}_{i+\frac{1}{2}}\}} (-1)^{\sum_{j=1}^{L}(s_{j-1}+s_j)\widehat{s}_{j-\frac{1}{2}}+\widehat{t}s_L} |\{\widehat{s}_{i+\frac{1}{2}}\}\rangle \,,$$

$$\mathcal{N}^\dagger |\{\widehat{s}_{i+\frac{1}{2}}\}\rangle = \frac{1}{2^{\frac{L}{2}}} \sum_{\{s_i\}} (-1)^{\sum_{j=1}^{L}(\widehat{s}_{j-\frac{1}{2}}+\widehat{s}_{j+\frac{1}{2}})s_j+t\widehat{s}_{\frac{1}{2}}} |\{s_i\}\rangle \,,$$
(2.4)

where we use $s_0 = s_L + t$, $\widehat{s}_{\frac{1}{2}} = \widehat{s}_{L+\frac{1}{2}} + \widehat{t}$ to identify the phases $\sum_{j=1}^{L}(s_{j-1} + s_j)\widehat{s}_{j-\frac{1}{2}} + \widehat{t}s_L = \sum_{j=1}^{L}(\widehat{s}_{j-\frac{1}{2}} + \widehat{s}_{j+\frac{1}{2}})s_j + t\widehat{s}_{\frac{1}{2}}$. The exponents in (2.4) are reminiscent of the minimal coupling of the gauge fields. The boundary terms in the exponents are chosen to give the correct mapping of symmetry-twist sectors. It is also straightforward to check that the duality operator is Hermitian, $\langle\{s_i\}|\mathcal{N}^\dagger|\{\widehat{s}_{i+\frac{1}{2}}\}\rangle = \langle\{\widehat{s}_{i+\frac{1}{2}}\}|\mathcal{N}|\{s_i\}\rangle$. By definition $\mathcal{N}$ commutes with the Hamiltonian (under suitable relabeling as shown below), hence $\mathcal{N}$ is mobile along the time direction.

**Mapping between Pauli operators:** KW transformation (2.4) induces the standard transformation of Pauli operators

$$\mathcal{N}\sigma_i^z \sigma_{i+1}^z |\psi\rangle = \widehat{\sigma}_{i+\frac{1}{2}}^x \mathcal{N} |\psi\rangle \,, \quad \mathcal{N}\sigma_i^x |\psi\rangle = \widehat{\sigma}_{i-\frac{1}{2}}^z \widehat{\sigma}_{i+\frac{1}{2}}^z \mathcal{N} |\psi\rangle \,, \quad \forall |\psi\rangle \in \mathcal{H} \,. \tag{2.5}$$

**Non-invertible fusion rules:** By acting the product of $\widehat{U} \times \mathcal{N}$, $\mathcal{N} \times U$ and $\mathcal{N}^\dagger \times \mathcal{N}$ on a general state $|\psi\rangle$, we find the following fusion rules:

$$\mathcal{N} \times U = (-1)^{\widehat{t}}\mathcal{N} \,, \quad \widehat{U} \times \mathcal{N} = (-1)^t \mathcal{N} \,, \quad \mathcal{N}^\dagger \times \mathcal{N} = 1 + (-1)^{\widehat{t}}U \,. \tag{2.6}$$

See [154, Sec. 2.2] for an explanation of the factors $(-1)^t$ and $(-1)^{\widehat{t}}$. In particular, the last fusion rule shows that the KW duality operator $\mathcal{N}$ is non-invertible.

**Mapping between symmetry-twist sectors:**   The symmetry and twist sectors before KW transformation $(u, t)$ and after KW transformation $(\widehat{u}, \widehat{t})$ are related by

$$u = \widehat{t}, \quad t = \widehat{u}. \tag{2.7}$$

**Example:**   A canonical example with a KW duality symmetry is the Ising model with a critical transverse field.

$$H_{\text{Ising}} = -\sum_{i=1}^{L} \sigma_i^z \sigma_{i+1}^z - \sum_{i=1}^{L} \sigma_i^x. \tag{2.8}$$

By using (2.5), one obtains the dual Hamiltonian

$$\widehat{H}_{\text{Ising}} = -\sum_{i=1}^{L} \widehat{\sigma}_{i+\frac{1}{2}}^x - \sum_{i=1}^{L} \widehat{\sigma}_{i-\frac{1}{2}}^z \widehat{\sigma}_{i+\frac{1}{2}}^z. \tag{2.9}$$

Schematically, after relabeling the spins on the sites and on the links, the two Hamiltonians coincide, $H_{\text{Ising}} = \widehat{H}_{\text{Ising}}$, hence the critical Ising model has a KW duality symmetry. We will make this statement more rigorous in Sec. 2.4 by performing the KW transformation on the same lattice.

## 2.3   KW duality defects

In this subsection, we discuss KW duality defects in Hamiltonian formalism. Although it has been widely discussed in the context of continuum field theories [55,106,107], an explicit realization on a lattice is much less well-known (most of what we present below can be found in [52,106,157]). To be self-contained, we will describe in detail the definition of the KW duality defect via the KW duality twist operator, their fusion rule, and their mobility along the space direction.

**KW duality defect from KW duality twist operator:**   Inserting a KW duality defect is equivalent to performing a KW transformation on the half-space. For simplicity, we only consider 1d infinite chains throughout this subsection. Inserting a KW duality defect at the origin amounts to performing a KW transformation on the left half of the lattice, or equivalently applying a KW duality twist operator $\mathcal{N}_0^t$ on the half, as shown in Fig. 2.

The KW duality twist operator terminating at $i = 0$ is defined as

$$\mathcal{N}_0^t \left| \{s_i\} \right\rangle = \frac{1}{2^{\ell/2}} \sum_{\{\widehat{s}_{i-\frac{1}{2}}\}_{i\leq 0}} (-1)^{\sum_{j\leq 0}(s_{j-1}+s_j)\widehat{s}_{j-\frac{1}{2}}} \left| \{\widehat{s}_{i-\frac{1}{2}}\}_{i\leq 0}; \{s_i\}_{i>0} \right\rangle, \tag{2.10}$$

which creates the KW duality defect at $i = \frac{1}{2}$. The resulting Hilbert space is defined on half-integer links to the left of the origin, and integer sites to the right, as shown in Fig. 2. The overall normalization descends from that in (2.4) where $\ell$ counts the number of sites covered by $\mathcal{N}_0^t$.

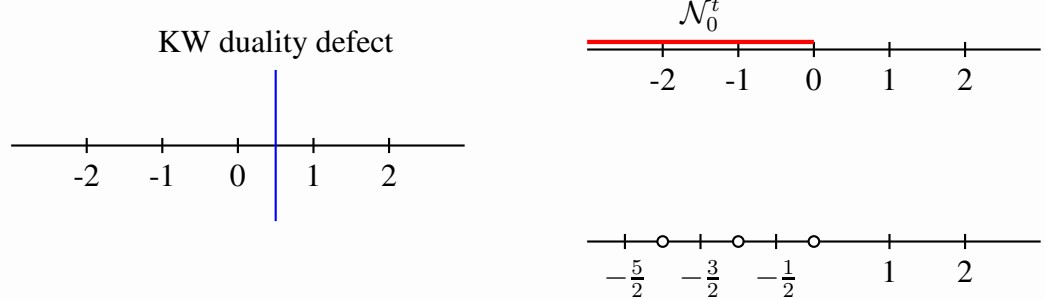

Figure 2: KW duality defects and twist operators. The left figure is the KW duality defect along the time direction, localized at $i = \frac{1}{2}$. Such a defect can be created, in the Hamiltonian formalism, by acting the original system with a KW duality twist operator $\mathcal{N}_0^t$, as shown in the right top figure. In resulting Hilbert space is represented in the right bottom figure, where the vertical slashes represent where the spins live, and they occupy half-integer links to the left of the origin and integer sites to the right. The empty circles are sites to the left of the origin where no spins are supported.

While $\ell \to \infty$ since $\mathcal{N}_0^t$ is infinitely long, we formally keep this normalization, which will be useful below. [9] It is also straightforward to obtain the action of $(\mathcal{N}_0^t)^\dagger$

$$(\mathcal{N}_0^t)^\dagger \, |\{\widehat{s}_{i-\frac{1}{2}}\}_{i\leq 0}; \{s_i\}_{i>0}\rangle = \frac{1}{2^{\frac{\ell}{2}}} \sum_{\{s_i\}_{i\leq 0}} (-1)^{\sum_{j\leq 0}(s_{j-1}+s_j)\widehat{s}_{j-\frac{1}{2}}} \, |\{s_i\}\rangle \, . \tag{2.11}$$

**Fusion rule:** We first derive the fusion rule of the KW duality defects. Fusing two KW duality defects on top of each other amounts to multiplying two KW duality twist operators at the same position. Since the Hilbert spaces before and after $\mathcal{N}_0^t$ are different, it is most convenient to compute the fusion $(\mathcal{N}_0^t)^\dagger \times \mathcal{N}_0^t$ [10]

$$(\mathcal{N}_0^t)^\dagger \times \mathcal{N}_0^t \, |\{s_i\}\rangle = \frac{1}{2^\ell} \sum_{\{\widehat{s}_{i-\frac{1}{2}}\}_{i\leq 0}, \{s_i'\}_{i\leq 0}} (-1)^{\sum_{j\leq 0}(s_{j-1}+s_j)\widehat{s}_{j-\frac{1}{2}} + \sum_{j\leq 0}(s_{j-1}'+s_j')\widehat{s}_{j-\frac{1}{2}}} \, |\{s_i'\}_{i\leq 0}; \{s_i\}_{i>0}\rangle$$

$$= (1 + U_0^t) \, |\{s_i\}\rangle \, . \tag{2.12}$$

Hence the fusion of the KW duality defect with its Hermitian conjugate is equivalent to the sum of an identity defect and a $\mathbb{Z}_2$ defect. This is consistent with the fusion rule among two KW duality operators in (2.6). [11]

---

[9] Another way to make sense of the normalization is to place the KW duality twist operator on a finite interval, with both left and right boundaries.

[10] To compute the fusion rule $\mathcal{N}_0^t \times \mathcal{N}_0^t$, one needs to identify the Hilbert spaces $|\{\widehat{s}_{i-\frac{1}{2}}\}_{i\leq 0}; \{s_i\}_{i>0}\rangle$ and $|\{s_i\}\rangle$ by shifting the link variables to sites.

[11] Because we are working on an infinite chain, we do not see the factor $(-1)^{\widehat{t}}$ here as opposed to (2.6).

Let us also check the fusion between the $\mathbb{Z}_2$ defect and the KW duality defect. Fusing the $\mathbb{Z}_2$ defect from the right gives

$$\mathcal{N}_0^t \times U_0^t \,|\{s_i\}\rangle = \frac{1}{2^{\ell/2}} \sum_{\{\widehat{s}_{i-\frac{1}{2}}\}_{i\leq 0}} (-1)^{\sum_{j\leq 0}(s_{j-1}+1+s_j+1)\widehat{s}_{j-\frac{1}{2}}} \,|\{\widehat{s}_{i-\frac{1}{2}}\}_{i\leq 0}; \{s_i\}_{i>0}\rangle$$
$$= \mathcal{N}_0^t \,|\{s_i\}\rangle \,. \tag{2.13}$$

This is also consistent with (2.6). Fusing the $\mathbb{Z}_2$ defect from the left yields

$$\widehat{U}_{-\frac{1}{2}}^t \times \mathcal{N}_0^t \,|\{s_i\}\rangle = \frac{1}{2^{\ell/2}} \sum_{\{\widehat{s}_{i-\frac{1}{2}}\}_{i\leq 0}} (-1)^{\sum_{j\leq 0}(s_{j-1}+s_j)(\widehat{s}_{j-\frac{1}{2}}+1)} \,|\{\widehat{s}_{i-\frac{1}{2}}\}_{i\leq 0}; \{s_i\}_{i>0}\rangle$$
$$= \mathcal{N}_0^t \sigma_0^z \,|\{s_i\}\rangle \,. \tag{2.14}$$

We thus naively arrive at $\widehat{U}_{-\frac{1}{2}}^t \times \mathcal{N}_0^t = \mathcal{N}_0^t \sigma_0^z$, where a local operator $\sigma_0^z$ is located at the defect locus. In the Hamiltonian formalism, note that two defects are equivalent to each other if they are related by a local unitary operator, see the discussions around (2.3) as well as [157, Sec. VI.A]. [12] Hence we identify the twist operator $\mathcal{N}_0^t \sigma_0^z$ with $\mathcal{N}_0^t$, under which we find the same fusion rule $\widehat{U}_{-\frac{1}{2}}^t \times \mathcal{N}_0^t = \mathcal{N}_0^t$ as in (2.6). In summary, the fusion rules involving the KW duality defects are

$$(\mathcal{N}_0^t)^\dagger \times \mathcal{N}_0^t = 1 + U_0^t, \quad \mathcal{N}_0^t \times U_0^t = \mathcal{N}_0^t, \quad \widehat{U}_{-\frac{1}{2}}^t \times \mathcal{N}_0^t = \mathcal{N}_0^t \,. \tag{2.15}$$

**Mobility of KW duality defects:** We proceed to probe the mobility of the KW duality defect, by shifting it along the space. Following the discussion around (2.3), we need to find a local unitary operator $W$ such that

$$\mathcal{N}_1^t = \mathcal{N}_0^t W \,. \tag{2.16}$$

Here, $\mathcal{N}_1^t$ is defined by a shift $\mathcal{N}_0^t$ by one site,

$$\mathcal{N}_1^t \,|\{s_i\}\rangle = \frac{1}{2^{\ell/2}} \sum_{\{\widehat{s}_{i-\frac{1}{2}}\}_{i\leq 1}} (-1)^{\sum_{j\leq 1}(s_{j-1}+s_j)\widehat{s}_{j-\frac{1}{2}}} \,|\{\widehat{s}_{i-\frac{1}{2}}\}_{i\leq 1}; \{s_i\}_{i>1}\rangle \,. \tag{2.17}$$

The form of $W$ has already been found in [52, Sec. 3.3] and [157, Sec. VI.A]. Concretely,

$$W = \mathsf{CZ}_{0,1}\mathsf{H}_1 \,, \tag{2.18}$$

where $\mathsf{CZ}_{0,1}$ is the control Z gate acting on site $0$ and $1$, and $\mathsf{H}_1$ is the Hadamard gate acting on site $1$. In terms of Pauli matrices, they are

$$\mathsf{CZ}_{0,1} = \frac{1}{2}(1 + \sigma_0^z + \sigma_1^z - \sigma_0^z\sigma_1^z), \quad \mathsf{H}_1 = \frac{1}{\sqrt{2}}(\sigma_1^z + \sigma_1^x) \,, \tag{2.19}$$

---

[12]Note that we only identify two defects that differ by a *local* unitary operator. Locality here means the unitary operator is of the same spatial dimension as the boundary of the twist operator, and in this case is zero-dimensional. We do not a priori identify $\widehat{U}_{-\frac{1}{2}}^t \times \mathcal{N}_0^t$ with $\mathcal{N}_0^t$ because $\widehat{U}_{-\frac{1}{2}}^t$ is not local, although they are eventually identified after computation.

from which we can derive their action on the basis product states,

$$\mathsf{CZ}_{0,1}\,|s_0, s_1\rangle = (-1)^{s_0 s_1}\,|s_0, s_1\rangle\,, \quad \mathsf{H}_1\,|s_1\rangle = \frac{1}{\sqrt{2}}\sum_{s_1'=0}^{1}(-1)^{s_1 s_1'}\,|s_1'\rangle\,. \tag{2.20}$$

The $\mathsf{CZ}_{0,1}$ operator maps $\sigma_0^x$ to $\sigma_0^x \sigma_1^z$, and leaves everything else invariant. The $\mathsf{H}_1$ operation exchanges $\sigma_1^x \leftrightarrow \sigma_1^z$. As a quick consistency check, note that $\mathcal{N}_i^t$ satisfies the fusion rule $(\mathcal{N}_i^t)^\dagger \times \mathcal{N}_i^t = 1 + U_i^t$ for $i = 0, 1$. Combining with (2.16), we have $U_1^t = W^\dagger U_0^t W$. Since $U_0^t$ is $\prod_{i \leq 0} \sigma_i^x$, first conjugating by $\mathsf{CZ}_{0,1}$ appends a $\sigma_1^z$ to $U_0^t$, and further conjugating by $\mathsf{H}_1$ maps $\sigma_1^z$ to $\sigma_1^x$, giving rise to $U_1^t$, as expected. With this, let us explicitly check (2.16),

$$\mathcal{N}_0^t W\,|\{s_i\}\rangle = \mathcal{N}_0^t \frac{1}{\sqrt{2}}\sum_{s_1'}(-1)^{s_1 s_1' + s_0 s_1'}\,|\{s_i\}_{i\leq 0}; s_1'; \{s_i\}_{i>1}\rangle$$

$$= \frac{1}{2^{(\ell+1)/2}}\sum_{\{\widehat{s}_{i-\frac{1}{2}}\}_{i\leq 0}, s_1'}(-1)^{\sum_{j\leq 0}(s_{j-1}+s_j)\widehat{s}_{j-\frac{1}{2}} + (s_0+s_1)s_1'}\,|\{\widehat{s}_{i-\frac{1}{2}}\}_{i\leq 0}; s_1'; \{s_i\}_{i>1}\rangle\,. \tag{2.21}$$

After relabeling the dummy variable $s_1' \to \widehat{s}_{\frac{1}{2}}$, the right-hand side in the second row is exactly the same as $\mathcal{N}_1^t\,|\{s_i\}\rangle$ in (2.17). This shows that the KW duality defect is mobile under translation along the space direction.

So far, we showed that the KW duality operator is mobile under translation along the time direction, and the KW duality defect is mobile under translation along the space direction. To fully establish the topologicalness of the KW duality symmetry generator, we also need to discuss the mobility when we bend the time-like defect into the space-like operator. See [83, 142–144] for related discussions on invertible and subsystem symmetries. We will not try to prove the full topologicalness of the KW duality symmetry generator in the current work.

## 2.4 KW duality symmetry

We further redefine the spins on the links to spins on sites. Concretely,

$$\widehat{s}_{i+\frac{1}{2}} \to s_i' \tag{2.22}$$

for any $i$. We also relabel $\widehat{t} \to t', \widehat{u} \to u'$. Under this modification, the KW duality operator maps within a single Hilbert space. In this way, the KW duality can be lifted as a symmetry of the lattice models. Under this redefinition, (2.4) becomes

$$\bar{\mathcal{N}}\,|\{s_i\}\rangle := \frac{1}{2^{\frac{L}{2}}}\sum_{\{s_i'\}}(-1)^{\sum_{j=1}^{L}(s_j+s_{j+1})s_j' + t's_{L+1}}\,|\{s_i'\}\rangle = \frac{1}{2^{\frac{L}{2}}}\sum_{\{s_i'\}}(-1)^{\sum_{j=1}^{L}(s_{j-1}'+s_j')s_j + ts_0'}\,|\{s_i'\}\rangle\,. \tag{2.23}$$

Here we use a bar to distinguish it from (2.4). KW transformation $\bar{\mathcal{N}}$ induces the following transformation of Pauli operators

$$\bar{\mathcal{N}}\sigma_i^z \sigma_{i+1}^z\,|\psi\rangle = \sigma_i^x \bar{\mathcal{N}}\,|\psi\rangle\,, \quad \bar{\mathcal{N}}\sigma_i^x\,|\psi\rangle = \sigma_{i-1}^z \sigma_i^z \bar{\mathcal{N}}\,|\psi\rangle\,, \quad \forall\,|\psi\rangle \in \mathcal{H}\,. \tag{2.24}$$

and therefore leaves the Ising model with a critical transverse field invariant. One advantage of this redefinition is that it now makes sense to compute $\bar{\mathcal{N}} \times \bar{\mathcal{N}}$, while before redefinition it only makes sense to compute $\mathcal{N} \times \mathcal{N}^\dagger$ or $\mathcal{N}^\dagger \times \mathcal{N}$. With this definition, it is straightforward to calculate the fusion of symmetry operator $\bar{\mathcal{N}}$:

$$\bar{\mathcal{N}} \times U = (-1)^{t'}\bar{\mathcal{N}}, \quad U \times \bar{\mathcal{N}} = (-1)^{t}\bar{\mathcal{N}}, \quad \bar{\mathcal{N}} \times \bar{\mathcal{N}} = (-1)^{tt'}(1 + (-1)^{t'}U)T, \qquad (2.25)$$

where $T$ is one-site translation operator

$$T\,|\{s_i\}\rangle = |\{s_i' = s_{i+1}\}\rangle \,. \qquad (2.26)$$

For example, we can confirm the last identity in (2.25) by the following calculation

$$
\begin{aligned}
\bar{\mathcal{N}}\bar{\mathcal{N}}\,|\{s_i\}\rangle &= \frac{1}{2^L} \sum_{\{s_i'\},\{s_i''\}} (-1)^{\sum_{j=1}^{L}(s_j+s_{j+1})s_j'+t's_{L+1}+\sum_{j=1}^{L}(s_{j-1}''+s_j'')s_j'+t's_0''}\,|\{s_i''\}\rangle \\
&= \sum_{\{s_i''\}} \left(\prod_{j=1}^{L} \delta_{s_j+s_{j+1}+s_{j-1}''+s_j''}\right)(-1)^{t'(s_{L+1}+s_0'')}\,|\{s_i''\}\rangle \\
&= \sum_{\{s_i''\}} \left(\prod_{j=1}^{L} \delta_{s_j+s_{j+1}+s_{j-1}''+s_j''}\right)(-1)^{t't+t'(s_1+s_0'')}\,|\{s_i''\}\rangle \\
&= (-1)^{tt'}\left(|\{s_i'' = s_{i+1}\}\rangle + (-1)^{t'}\,|\{s_i'' = s_{i+1}+1\}\rangle\right)
\end{aligned}
\qquad (2.27)
$$

Under (2.22), the twist operator (2.10) is modified to

$$
\begin{aligned}
\bar{\mathcal{N}}_0^t\,|\{s_i\}\rangle &= \frac{1}{2^{\ell/2}} \sum_{\{s_i'\}_{i\leq 0}} (-1)^{\sum_{j\leq 0}(s_j+s_{j+1})s_j'}\,|\{s_i'\}_{i\leq 0}; \{s_i\}_{i>0}\rangle \\
&= \frac{1}{2^{\ell/2}} \sum_{\{s_i'\}_{i\leq 0}} (-1)^{\sum_{j\leq 0}(s_{j-1}'+s_j')s_j+s_0's_1}\,|\{s_i'\}_{i\leq 0}; \{s_i\}_{i>0}\rangle
\end{aligned}
\qquad (2.28)
$$

which creates the corresponding defect at $i = 0$. One can directly calculate the fusion rules of defects. For example,

$$
\begin{aligned}
\bar{\mathcal{N}}_0^t \times \bar{\mathcal{N}}_0^t\,|\{s_i\}\rangle &= \frac{1}{2^\ell} \sum_{\{s_i''\}_{i\leq 0},\{s_i'\}_{i\leq 0}} (-1)^{\sum_{j\leq 0}(s_j+s_{j+1})s_j'+\sum_{j\leq 0}(s_{j-1}''+s_j'')s_j'+s_0''s_1}\,|\{s_i''\}_{i\leq 0}; \{s_i\}_{i>0}\rangle \\
&= \left(\prod_{j\leq 1} \delta_{s_j+s_{j-1}''} + \prod_{j\leq 1} \delta_{s_j+s_{j-1}''+1}\right)(-1)^{s_0''s_1}\,|\{s_i''\}_{i\leq 0}; \{s_i\}_{i>0}\rangle \\
&= (-1)^{s_1}\,|\{s_i'' = s_{i+1}\}_{i\leq 0}; \{s_i\}_{i>0}\rangle + |\{s_i'' = s_{i+1}+1\}_{i\leq 0}; \{s_i\}_{i>0}\rangle \\
&= (\sigma_0^z + U_0^t)T_0^t\,|\{s_i\}\rangle \,,
\end{aligned}
\qquad (2.29)
$$

where $T_0^t$ is the translation twist operator at $i = 0$ and shifts the left half chain by one site

$$T_0^t\,|\{s_i\}\rangle = |\{s_i' = s_{i+1}\}_{i\leq 0}; \{s_i\}_{i>0}\rangle \,. \qquad (2.30)$$

We can identify $\sigma_0^z T_0^t$ and $T_0^t$ in the fusion rule because they are related by the local unitary operator $\sigma_0^z$. In summary, we have

$$\bar{\mathcal{N}}_0^t \times U_0^t = U_0^t \times \bar{\mathcal{N}}_0^t = \bar{\mathcal{N}}_0^t, \quad \bar{\mathcal{N}}_0^t \times \bar{\mathcal{N}}_0^t = (1 + U_0^t)T_0^t. \tag{2.31}$$

The symmetry defect $\bar{\mathcal{N}}_0^t$ is movable because when acting the local operator $W' = \mathsf{CZ}_{1,2}\mathsf{H}_1$ we will get the defect $\bar{\mathcal{N}}_1^t = \bar{\mathcal{N}}_0^t W'$ acting at one site right.[13]

## 2.5   Anomaly of KW duality symmetry

We end this section by commenting on the anomaly of the KW duality symmetry. The anomaly of non-invertible symmetries in $(1+1)$d has been discussed in [55, 106] via modular transformation, and was later more systematically explored in [56, 59]. In this subsection, we review the approach in [106] in probing the anomaly of the KW duality symmetry, as a preparation for the anomaly of the subsystem KW symmetry.

The anomaly of the KW duality symmetry can be seen by contradiction. Suppose it is non-anomalous, which means that there is an invertible theory (i.e. a gapped theory with one ground state) with KW duality symmetry. The only invertible theory is a trivial theory, and in the fixed point limit, the partition function is $Z_{\text{triv}} = 1$. Performing the KW duality transformation amounts to gauging the $\mathbb{Z}_2$ symmetry, which maps $Z_{\text{triv}} = 1$ to a $\mathbb{Z}_2$ symmetry breaking phase [14]

$$Z_{\text{SSB}}[A] = \sum_{a \in \mathbb{Z}_2} (-1)^{\int aA} = \sum_{W_\tau^a, W_x^a = 0,1} (-1)^{W_\tau^a W_x^A + W_x^a W_\tau^A} = \delta(W_\tau^A)\delta(W_x^A). \tag{2.33}$$

where in the second equality we assume the spacetime to be a torus, and $W_{\tau,x}^{a,A}$ is the Wilson line of $a, A$ along the $\tau, x$ direction respectively. Clearly, a $\mathbb{Z}_2$ SSB theory is not equivalent to a trivial theory. This means that the trivial theory is not compatible with the KW duality symmetry, i.e. the KW duality symmetry is anomalous.

---

[13]The translation defect in the fusion rule of two KW operators/defects in the transverse field Ising model was emphasized in [153], which is termed non-invertible translation symmetry. This translation operator/defect could not be seen if one compute $\mathcal{N}^\dagger \times \mathcal{N}$ when we define the Hilbert space after KW on links rather than on sites. It is interesting to note that the same translation effect has been implicitly observed in [157, Sec. VI]. The authors of [157] discussed the critical transverse field Ising model with two defects on the closed chain. When the two defects fuses, e.g. at the fourth site, the corresponding Hamiltonian with $L$ spins is

$$H_{D_\sigma \times D_\sigma} = -\left( \sum_{i \neq 3,4} \sigma_i^z \sigma_{i+1}^z + \sum_{i \neq 4} \sigma_i^x + \sigma_3^z \sigma_4^x \sigma_5^x \right) \tag{2.32}$$

Since $\sigma_4^x$ commute with this Hamiltonian, the Hibert space can be decomposed into two sectors based on the parity of the fourth spin. The Hamiltonian in sector labeled by $\sigma^x = \pm 1$ is just the usual Ising chain with $L - 1$ spins under periodic boundary condition and twisted boundary condition. The decoupling effectively reduces the length of the spin chain by one, i.e. from $L$ spins to $L - 1$ spins, which is equivalent to adding a defect of translation.

[14]We will suppress the normalization.

# 3 Subsystem Kramers-Wannier duality operators and defects in $(2+1)$d spin systems

In this section, we generalize the discussion in Sec. 2 to lattice models with non-anomalous subsystem $\mathbb{Z}_2$ symmetries in $(2+1)$d. We also assume that the system is invariant under the gauging of the subsystem $\mathbb{Z}_2$ symmetry, which generalizes the KW duality symmetry to the subsystem KW duality symmetry. We further study various properties of its associated symmetry operators and defects, including their mobility, fusion rules, interplay with boundary conditions, and anomalies. Unlike in $(1+1)$d where the KW duality operator and KW duality defect share essentially the same properties, here we find that the subsystem KW duality operator and subsystem KW duality defect have very different properties. This is expected since theories with subsystem symmetry are incompatible with Lorentz symmetry.

## 3.1 Subsystem $\mathbb{Z}_2$ symmetry and twist operators

We first review the properties of a subsystem $\mathbb{Z}_2$ symmetry in $(2+1)$d, following [89]. Consider a non-anomalous (on-site) subsystem $\mathbb{Z}_2$ symmetric theory on either a square lattice with $L_x \times L_y$ sites or an infinite square lattice. On each site there is a spin-$\frac{1}{2}$ variable $s_{i,j} \in \{0, 1\}$. When the square lattice is infinitely large, the subsystem $\mathbb{Z}_2$ symmetry is generated by $\mathbb{Z}_2$ operators on each line and column

$$U_j^x = \prod_{i' \in \mathbb{Z}} \sigma_{i',j}^x, \quad U_i^y = \prod_{j' \in \mathbb{Z}} \sigma_{i,j'}^x, \quad i, j \in \mathbb{Z}, \tag{3.1}$$

whose eigenvalues we denote as $(-1)^{u_j^x}, (-1)^{u_i^y}$ respectively. Similar to the case of an ordinary $\mathbb{Z}_2$ symmetry in $(1+1)$d, the subsystem $\mathbb{Z}_2$ symmetry operators have $\mathbb{Z}_2$ self fusion rules and they are mobile/topological along the time direction. However, they are *not* mobile/topological in the space direction. We should label these operators by their locations on the lattice.

On a finite lattice (i.e. a torus), there is a constraint in the product of all symmetry generators

$$\prod_{i=1}^{L_x} U_i^y \prod_{j=1}^{L_y} U_j^x = 1, \tag{3.2}$$

since Pauli operators on each site appear twice on the left-hand side. The total number of symmetry generators is $2^{L_x + L_y - 1}$. The action of symmetry generators on a finite lattice is shown schematically in Fig. 3. We define the subsystem $\mathbb{Z}_2$ symmetry group as $\mathbb{Z}_2^{\text{sub}}$,

$$\mathbb{Z}_2^{\text{sub}} = \left\langle U_i^y, U_j^x \,\middle|\, (U_j^x)^2 = (U_i^y)^2 = \prod_{i=1}^{L_x} U_i^y \prod_{j=1}^{L_y} U_j^x = 1, \text{with } i = 1, ..., L_x, j = 1, ..., L_y \right\rangle. \tag{3.3}$$

We proceed to consider subsystem $\mathbb{Z}_2$ defects at each site along time direction. On an infinite square lattice, these defects can be created by acting twist operators $U_{0,j}^{xt}$ and $U_{i,0}^{yt}$ on the Hilbert

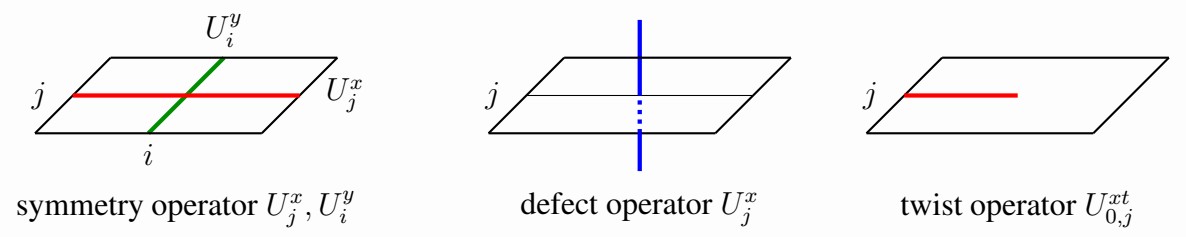

symmetry operator $U_j^x, U_i^y$      defect operator $U_j^x$      twist operator $U_{0,j}^{xt}$

Figure 3: Examples of subsystem $\mathbb{Z}_2$ symmetry operators, defect operators and twist operators.

space, where

$$U_{0,j}^{xt} = \prod_{i' \leq 0} \sigma_{i',j}^x, \quad U_{i,0}^{yt} = \prod_{j' \leq 0} \sigma_{i,j'}^x. \tag{3.4}$$

Clearly, the subsystem $\mathbb{Z}_2$ defects satisfy the $\mathbb{Z}_2$ fusion rule, $U_{0,j}^{xt} \times U_{0,j}^{xt} = U_{i,0}^{yt} \times U_{i,0}^{yt} = 1$. To see the mobility, note that $U_{1,j}^{xt}$ and $U_{0,j}^{xt}$ differ by a local unitary operator, i.e. $U_{1,j}^{xt} = U_{0,j}^{xt} \sigma_{1,j}^x$, hence the defect created by $U_{0,j}^{xt}$ is mobile along the $x$ direction. On the other hand, since $U_{0,j+1}^{xt}$ and $U_{0,j}^{xt}$ differ by a string (non-local) unitary operator, i.e. $U_{0,j+1}^{xt} = U_{0,j}^{xt} \prod_{i \leq 0} \sigma_{i,j}^x \sigma_{i,j+1}^x$, the defect is not mobile along the $y$ direction. Indeed, the defects created by $U_{0,j+1}^{xt}$ and $U_{0,j}^{xt}$ are regarded as inequivalent subsystem $\mathbb{Z}_2$ defects. The same discussion applies to $U_{i,0}^{yt}$ as well. We show some examples of symmetry operators, defect operators and twist operators diagrammatically in Fig. 3.

If we consider a finite square lattice instead, inserting a defect changes the boundary condition. The boundary condition is specified by [89]

$$|s_{i+L_x,j}\rangle = |s_{i,j} + t_j^x\rangle, \quad |s_{i,j+L_y}\rangle = |s_{i,j} + t_i^y\rangle, \quad |s_{i+L_x,j+L_y}\rangle = |s_{i,j} + t^{xy} + t_j^x + t_i^y\rangle, \tag{3.5}$$

where $t_j^x, t_i^y, t^{xy} \in \{0,1\}, \forall i,j$ label the twisted boundary conditions along the $j$-th row, $i$-th column and at the corner respectively. We further define boundary conditions of twist variables

$$t_{j+L}^x = t_j^x + t^{xy}, \quad t_{i+L_x}^y = t_i^y + t^{xy}. \tag{3.6}$$

There are $L_x + L_y + 1$ twist parameters but the Hamiltonian with subsystem $\mathbb{Z}_2$ symmetry depends only on the combinations $\{\mathfrak{t}_{j+\frac{1}{2}}^x := t_j^x + t_{j+1}^x, \mathfrak{t}_{i+\frac{1}{2}}^y := t_i^y + t_{i+1}^y\}$ [89]. Taking account of the constraint

$$\sum_{j=1}^{L_y} \mathfrak{t}_{j+\frac{1}{2}}^x = \sum_{i=1}^{L_x} \mathfrak{t}_{i+\frac{1}{2}}^y = t^{xy}, \tag{3.7}$$

only $L_x + L_y - 1$ twist variables are distinct.

Similar to the case of $(1+1)$d, the Hilbert space will be divided into symmetry-twist sectors labeled by eigenvalues of the symmetry generators and boundary conditions. Denote the eigenvalues of $U_j^x, U_i^y$ as $(-1)^{u_j^x}, (-1)^{u_i^y}$ respectively, with $u_j^x, u_i^y \in \{0,1\}$. The constraint on the symmetry generators (3.2) leads to $L_x + L_y - 1$ independent symmetry generators, dividing the entire Hilbert space (with a fixed boundary condition) to $2^{L_x+L_y-1}$ sectors. Similarly, the $L_x + L_y - 1$

distinct twist variables $(\mathfrak{t}^x_{j+\frac{1}{2}}, \mathfrak{t}^y_{i+\frac{1}{2}})$ further divide each symmetry sector into $2^{L_x+L_y-1}$ parts. In total, there are $4^{L_x+L_y-1}$ distinguished symmetry-twist sectors labeled by $(u^x_j, u^y_i, \mathfrak{t}^x_{j+\frac{1}{2}}, \mathfrak{t}^y_{i+\frac{1}{2}})$.

## 3.2 Subsystem KW duality operators

In this subsection, we generalize the KW transformation in $(1+1)$d to the *subsystem KW transformation*, defined by gauging the entire subsystem $\mathbb{Z}_2$ symmetry in $(2+1)$d. We also assume the theory to be invariant under the subsystem KW transformation. We will demonstrate the construction of the codimension one surface duality operator $\mathcal{N}^{\text{sub}}$ acting on the Hilbert space implementing the subsystem KW transformation and derive its fusion rule. Additional technical details about turning on background gauge fields are included in App. A.

**Dual lattice:** Subsystem KW transformation maps the original square lattice with spin $\{s_{i,j}\}$ to the dual lattice with spin $\{\widehat{s}_{i+\frac{1}{2},j+\frac{1}{2}}\}$. Each spin $\widehat{s}_{i+\frac{1}{2},j+\frac{1}{2}} \in \{0,1\}$ lives on the plaquette with boundary conditions

$$
\begin{aligned}
|\widehat{s}_{i+\frac{1}{2}+L_x,j+\frac{1}{2}}\rangle &= |\widehat{s}_{i+\frac{1}{2},j+\frac{1}{2}} + \widehat{t}^x_{j+\frac{1}{2}}\rangle, \\
|\widehat{s}_{i+\frac{1}{2},j+\frac{1}{2}+L_y}\rangle &= |\widehat{s}_{i+\frac{1}{2},j+\frac{1}{2}} + \widehat{t}^y_{i+\frac{1}{2}}\rangle, \\
|\widehat{s}_{i+\frac{1}{2}+L_x,j+\frac{1}{2}+L_y}\rangle &= |\widehat{s}_{i+\frac{1}{2},j+\frac{1}{2}} + \widehat{t}^{xy} + \widehat{t}^x_{j+\frac{1}{2}} + \widehat{t}^y_{i+\frac{1}{2}}\rangle,
\end{aligned}
\tag{3.8}
$$

where

$$
\widehat{t}^x_{j+\frac{1}{2}+L_y} = \widehat{t}^x_{j+\frac{1}{2}} + \widehat{t}^{xy}, \quad \widehat{t}^y_{i+\frac{1}{2}+L_x} = \widehat{t}^y_{i+\frac{1}{2}} + \widehat{t}^{xy}, \quad \widehat{t}^x_{j+\frac{1}{2}}, \widehat{t}^y_{i+\frac{1}{2}}, \widehat{t}^{xy} \in \{0,1\}, \forall i,j.
\tag{3.9}
$$

We use combined boundary conditions

$$
\widehat{\mathfrak{t}}^x_j := \widehat{t}^x_{j-\frac{1}{2}} + \widehat{t}^x_{j+\frac{1}{2}}, \quad \widehat{\mathfrak{t}}^y_i := \widehat{t}^y_{i-\frac{1}{2}} + \widehat{t}^y_{i+\frac{1}{2}}.
\tag{3.10}
$$

Among them, there are only $L_x + L_y - 1$ independent variables.

Let the Pauli operator $\tau^z_{i+\frac{1}{2},j+\frac{1}{2}}, \tau^x_{i+\frac{1}{2},j+\frac{1}{2}}$ act on the dual lattice. The new subsystem $\mathbb{Z}_2$ symmetry is generated by

$$
\widehat{U}^x_{j+\frac{1}{2}} = \prod_{i=1}^{L_x} \tau^x_{i+\frac{1}{2},j+\frac{1}{2}}, \quad \widehat{U}^y_{i+\frac{1}{2}} = \prod_{j=1}^{L_y} \tau^x_{i+\frac{1}{2},j+\frac{1}{2}},
\tag{3.11}
$$

with the constraint $\prod_{j=1}^{L_y} \widehat{U}^x_{j+\frac{1}{2}} \prod_{i=1}^{L_x} \widehat{U}^y_{i+\frac{1}{2}} = 1$. Denoting the eigenvalues of the symmetry generators as $(-1)^{\widehat{u}^x_{j+\frac{1}{2}}}, (-1)^{\widehat{u}^y_{i+\frac{1}{2}}} = \pm 1$, the Hilbert space on the dual lattice is divided into $2^{L_x+L_y-1} \times 2^{L_x+L_y-1}$ symmetry-twist sectors labeled by $(\widehat{u}^x_{j+\frac{1}{2}}, \widehat{u}^y_{i+\frac{1}{2}}, \widehat{\mathfrak{t}}^x_j, \widehat{\mathfrak{t}}^y_i)$.

**Subsystem KW transformation:** We use short-hand notations $|\{s_{i,j}\}\rangle$ and $|\{\widehat{s}_{i+\frac{1}{2},j+\frac{1}{2}}\}\rangle$ for basis states. Since the action of a symmetry generator in (3.1) only changes the spin along a line or a column, we use $|\{s_{i,j}\};\{s'_{i,j'}\}\rangle$ for action on $j'$-th line or $|\{s_{i,j}\};\{s'_{i',j}\}\rangle$ for action on $i'$-th column. We adopt the same notation for action on the dual Hilbert space. For instance, the symmetry generators $U_j^x, U_i^y, \widehat{U}_{j+\frac{1}{2}}^x, \widehat{U}_{i+\frac{1}{2}}^y$ act as

$$U_{j'}^x |\{s_{i,j}\}\rangle = |\{s_{i,j}\};\{1 - s_{i,j'}\}\rangle, \quad \widehat{U}_{j'+\frac{1}{2}}^x |\{\widehat{s}_{i+\frac{1}{2},j+\frac{1}{2}}\}\rangle = |\{\widehat{s}_{i+\frac{1}{2},j+\frac{1}{2}}\};\{1 - \widehat{s}_{i+\frac{1}{2},j'+\frac{1}{2}}\}\rangle \ . \tag{3.12}$$

Compared with $(1+1)$d, we define the action of subsystem KW transformation $\mathcal{N}^{\text{sub}}$ on the entire Hilbert space by attaching a phase to the dual basis state and summing over all the dual basis states. The exponent of the phase contains the term reminiscent of the minimal coupling of a subsystem gauge field on the plaquette as well as the boundary term that gives the following mapping between symmetry-twist sectors

$$\widehat{u}_{j+\frac{1}{2}}^x = \mathfrak{t}_{j+\frac{1}{2}}^x, \quad \widehat{u}_{i+\frac{1}{2}}^y = \mathfrak{t}_{i+\frac{1}{2}}^y, \quad \widehat{\mathfrak{t}}_j^x = u_j^x, \quad \widehat{\mathfrak{t}}_i^y = u_i^y, \tag{3.13}$$

as shown in Fig. 4.

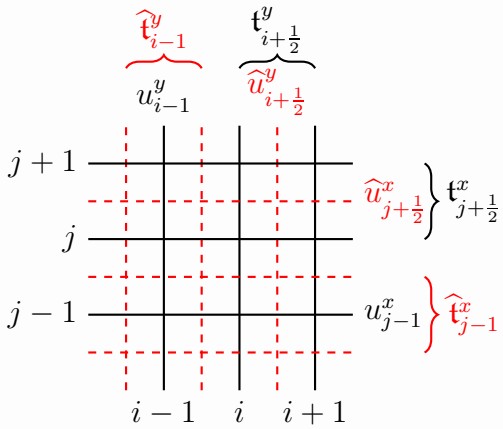

Figure 4: Mapping of symmetry-twist sectors.

Concretely, the subsystem KW duality operator is defined as

$$\mathcal{N}^{\text{sub}} |\{s_{i,j}\}\rangle = \frac{1}{2^{(L_x+L_y)/2}} \sum_{\{\widehat{s}_{i+\frac{1}{2},j+\frac{1}{2}}\}} (-1)^{C_{\text{bulk}}+C_{\text{bdy}}} |\{\widehat{s}_{i+\frac{1}{2},j+\frac{1}{2}}\}\rangle \ , \tag{3.14}$$

where in the exponent $C_{\text{bulk}}$ is the bulk minimal coupling between the original spins on sites and dual spins on the plaquette, and $C_{\text{bdy}}$ is the boundary coupling ensuring the correct symmetry-twist

sector transformation (3.13),

$$C_{\text{bulk}} := \sum_{i=1}^{L_x} \sum_{j=1}^{L_y} (s_{i-1,j-1} + s_{i,j-1} + s_{i-1,j} + s_{i,j})\widehat{s}_{i-\frac{1}{2},j-\frac{1}{2}},$$

$$C_{\text{bdy}} := \sum_{j=1}^{L_y} \widehat{t}^x_{j-\frac{1}{2}}(s_{L_x,j} + s_{L_x,j-1}) + \sum_{i=1}^{L_x} \widehat{t}^y_{i-\frac{1}{2}}(s_{i,L_y} + s_{i-1,L_y}) + \widehat{t}^{xy} s_{L_x,L_y},$$

(3.15)

where the spin with index 0 equal to the one with $L_x/L_y$ shifted by proper boundary condition, e.g. $s_{0,0} = s_{L_x,L_y} + t^{xy} + t^x_{L_y} + t^y_{L_x}$. As in $(1+1)$d, we adopt the same phase for the definition of the subsystem KW transformation $\mathcal{N}^{\text{sub}}$ acting on the dual Hilbert space $\widehat{\mathcal{H}}$. The subsystem KW transformation $\mathcal{N}^{\text{sub}}$ is Hermitian.

**Mapping between Pauli operators:**   From (3.14) and (3.15), the Pauli operators transform as

$$\mathcal{N}^{\text{sub}} \sigma^z_{i,j} \sigma^z_{i+1,j} \sigma^z_{i,j+1} \sigma^z_{i+1,j+1} |\psi\rangle = \widehat{\sigma}^x_{i+\frac{1}{2},j+\frac{1}{2}} \mathcal{N}^{\text{sub}} |\psi\rangle, \quad \forall |\psi\rangle \in \mathcal{H},$$

$$\mathcal{N}^{\text{sub}} \sigma^x_{i,j} |\psi\rangle = \widehat{\sigma}^z_{i-\frac{1}{2},j-\frac{1}{2}} \widehat{\sigma}^z_{i-\frac{1}{2},j+\frac{1}{2}} \widehat{\sigma}^z_{i+\frac{1}{2},j-\frac{1}{2}} \widehat{\sigma}^z_{i+\frac{1}{2},j+\frac{1}{2}} \mathcal{N}^{\text{sub}} |\psi\rangle, \quad \forall |\psi\rangle \in \mathcal{H}.$$

(3.16)

which is a natural generalization of the ordinary KW transformation in $(1+1)$d.

**Fusion rules:**   Fusion rules can be derived straightforwardly by acting the symmetry operators $U^x_j, U^y_i, \widehat{U}^x_{j+\frac{1}{2}}, \widehat{U}^y_{i+\frac{1}{2}}$ and the duality operator $\mathcal{N}^{\text{sub}}$ on a general state $|\psi\rangle = \sum_{\{s_{i,j}\}} \psi_{\{s_{i,j}\}} |\{s_{i,j}\}\rangle$, where $\psi_{\{s_{i,j}\}} = \langle \{s_{i,j}\}|\psi\rangle$ is the wavefunction coefficient. For example, acting $\mathcal{N}^{\text{sub}} \times U^x_{j'}$ on $|\psi\rangle$

$$\mathcal{N}^{\text{sub}} \times U^x_{j'} |\psi\rangle = \sum_{\{s_{i,j}\},\{\widehat{s}_i\}} \psi_{\{s_{i,j}\};\{1-s_{i,j'}\}} \left( \mathcal{N}^{\text{sub}} |\{s_{i,j}\}\rangle \right),$$

(3.17)

where we use redefinition of spin $s_{i,j'} \to 1 - s_{i,j'}$ in the $j'$-th line. Then we perform the subsystem KW transformation and redefine spins to send $\psi_{\{s_{i,j}\};\{1-s_{i,j'}\}}$ back to $\psi_{\{s_{i,j}\}}$. Under redefinition, the changes in the bulk minimal coupling terms cancel because spins with index $j'$ always appear in pairs. The only contribution is from the boundary term

$$\widehat{t}^x_{j'-\frac{1}{2}}(s_{L_x,j'} + s_{L_x,j'-1}) + \widehat{t}^x_{j'+\frac{1}{2}}(s_{L_x,j'+1} + s_{L_x,j'})$$

$$\to \widehat{t}^x_{j'-\frac{1}{2}}(1 - s_{L_x,j'} + s_{L_x,j'-1}) + \widehat{t}^x_{j'+\frac{1}{2}}(s_{L_x,j'+1} + 1 - s_{L_x,j'}),$$

(3.18)

leading to an extra factor $(-1)^{\widehat{t}^x_{j'-\frac{1}{2}} + \widehat{t}^x_{j'+\frac{1}{2}}} = (-1)^{\widehat{t}^x_{j'}}$. Therefore, the fusion rule is

$$\mathcal{N}^{\text{sub}} \times U^x_{j'} = (-1)^{\widehat{t}^x_{j'}} \mathcal{N}^{\text{sub}}.$$

(3.19)

By a similar argument, one can work out fusions with other symmetry generators. Here we only list the results:

$$\widehat{U}^x_{j'+\frac{1}{2}} \times \mathcal{N}^{\text{sub}} = (-1)^{t^x_{j'+\frac{1}{2}}} \mathcal{N}^{\text{sub}}, \quad \mathcal{N}^{\text{sub}} \times U^y_{i'} = (-1)^{\widehat{t}^y_{i'}} \mathcal{N}^{\text{sub}}, \quad \widehat{U}^y_{i'+\frac{1}{2}} \times \mathcal{N}^{\text{sub}} = (-1)^{t^y_{i'+\frac{1}{2}}} \mathcal{N}^{\text{sub}}.$$

(3.20)

For self fusion of the duality operator, by acting $(\mathcal{N}^{\text{sub}})^\dagger \times \mathcal{N}^{\text{sub}}$ on a general state $|\psi\rangle$ we get two copies of the phase

$$\sum_{i=1}^{L_x}\sum_{j=1}^{L_y}(s_{i-1,j-1}+s_{i,j-1}+s_{i-1,j}+s_{i,j}+s'_{i-1,j-1}+s'_{i,j-1}+s'_{i-1,j}+s'_{i,j})\widehat{s}_{i-\frac{1}{2},j-\frac{1}{2}}$$

$$+\sum_{j=1}^{L_y}\widehat{t}^x_{j-\frac{1}{2}}(s_{L_x,j}+s_{L_x,j-1}+s'_{L_x,j}+s'_{L_x,j-1})+\sum_{i=1}^{L_x}\widehat{t}^y_{i-\frac{1}{2}}(s_{i,L_y}+s_{i-1,L_y}+s'_{i,L_y}+s'_{i-1,L_y})$$

$$+\widehat{t}^{xy}(s_{L_x,L_y}+s'_{L_x,L_y})\,.$$

$$(3.21)$$

We then sum over states in the dual Hilbert space to get a product of delta function constraints on states in the original Hilbert space $\mathcal{H}$

$$s_{i-1,j-1}+s_{i,j-1}+s_{i-1,j}+s_{i,j}+s'_{i-1,j-1}+s'_{i,j-1}+s'_{i-1,j}+s'_{i,j}=0,\quad \forall i,j\,. \qquad (3.22)$$

The general solutions of the constraints are

$$s'_{i,j}=s_{i,j}+m^y_i+m^x_j,\quad m^y_i,m^x_j\in\{0,1\},\quad \forall i,j\,, \qquad (3.23)$$

where $m^y_i,m^x_j$ having value 1 means one insertion of $U^y_i,U^x_j$ respectively. We should then sum over all distinct solutions, [15]

$$M=\big\langle (m^y_i,m^x_j)\big|\, m^y_i,m^x_j\in\{0,1\},(m^y_i,m^x_j)\simeq(m^y_i+1,m^x_j+1)\big\rangle,\quad \forall i,j\,. \qquad (3.24)$$

The final fusion rule is

$$(\mathcal{N}^{\text{sub}})^\dagger \times \mathcal{N}^{\text{sub}}=\sum_{(m^y_i,m^x_j)\in M}(-1)^{\sum_{j=1}^{L_y}\widehat{t}^x_j m^x_j+\sum_{i=1}^{L_x}\widehat{t}^y_i m^y_i}\prod_{i=1}^{L_x}(U^y_i)^{m^y_i}\prod_{j=1}^{L_y}(U^x_j)^{m^x_j}\,, \qquad (3.25)$$

generalizing the ordinary Ising fusion rule. The right-hand side is a sum of the generators of $\mathbb{Z}_2^{\text{sub}}$, reminiscent of the condensation defect that appears in the fusion rule of duality defects in $(3+1)$d gauge theories [106,107,110,114,116,118]. As the operators $U^x_j$ and $U^y_i$ have restricted mobility, and they form grids, we denote the right-hand side as the *grid operator*,

$$\mathsf{Grid}_{\{\widehat{t}^y_i,\widehat{t}^x_j\}}=\sum_{(m^y_i,m^x_j)\in M}(-1)^{\sum_{j=1}^{L_y}\widehat{t}^x_j m^x_j+\sum_{i=1}^{L_x}\widehat{t}^y_i m^y_i}\prod_{i=1}^{L_x}(U^y_i)^{m^y_i}\prod_{j=1}^{L_y}(U^x_j)^{m^x_j} \qquad (3.26)$$

$$=\frac{1}{2}\prod_{i=1}^{L_x}\left(1+(-1)^{\widehat{t}^y_i}U^y_i\right)\prod_{j=1}^{L_y}\left(1+(-1)^{\widehat{t}^x_j}U^x_j\right), \qquad (3.27)$$

so that

$$(\mathcal{N}^{\text{sub}})^\dagger \times \mathcal{N}^{\text{sub}}=\mathsf{Grid}_{\{\widehat{t}^y_i,\widehat{t}^x_j\}}\,. \qquad (3.28)$$

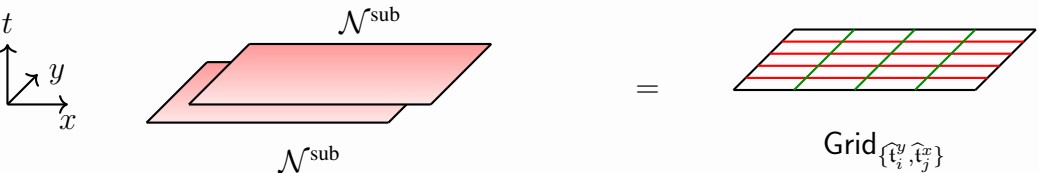

Figure 5: Fusion between two subsystem KW duality operators gives rise to a grid operator, where the grid is along the space direction.

We also present this fusion rule in Fig. 5.

To the authors' knowledge, the subsystem KW duality operator $\mathcal{N}^{\text{sub}}$ is the first explicit example of a *non-invertible* operator in models with subsystem symmetry, which generates a *non-invertible subsystem KW duality symmetry*.

**Example:** A canonical example with subsystem KW duality is the plaquette Ising model with the critical transverse field in $(2+1)$d:

$$H_{\text{PIsing}} = -\sum_{i=1}^{L_x}\sum_{j=1}^{L_y} \sigma^z_{i,j}\sigma^z_{i+1,j}\sigma^z_{i,j+1}\sigma^z_{i+1,j+1} - \sum_{i=1}^{L_x}\sum_{j=1}^{L_y} \sigma^x_{i,j}. \tag{3.29}$$

This model can describe a square array of superconductor grains with frustrating geometric phases and the states with eigenvalue $\sigma^z = \pm 1$ are associated with $p \pm ip$ order of each grain [171, 172]. The first term of Hamiltonian represents the phase acquired by a Cooper pair in the process of encircling a plaquette, while the second term arises from tunneling between the $p \pm ip$ order parameters. Applying the subsystem KW transformation, one obtains the dual Hamiltonian

$$\widehat{H}_{\text{PIsing}} = -\sum_{i=1}^{L_x}\sum_{j=1}^{L_y} \widehat{\sigma}^x_{i+\frac{1}{2},j+\frac{1}{2}} - \sum_{i=1}^{L_x}\sum_{j=1}^{L_y} \widehat{\sigma}^z_{i-\frac{1}{2},j-\frac{1}{2}}\widehat{\sigma}^z_{i+\frac{1}{2},j-\frac{1}{2}}\widehat{\sigma}^z_{i-\frac{1}{2},j+\frac{1}{2}}\widehat{\sigma}^z_{i+\frac{1}{2},j+\frac{1}{2}}. \tag{3.30}$$

After relabeling the spins on the sites and on the plaquettes, the two Hamiltonians coincide, $H_{\text{PIsing}} = \widehat{H}_{\text{PIsing}}$, hence the critical Plaquette Ising model has a subsystem KW duality symmetry. We will also formulate this symmetry by performing the subsystem KW transformation on the same lattice, as shown in Sec. 3.4.

Moreover, the plaquette Ising model (3.29), as a $(2+1)$d quantum spin model, has a corresponding 3d anisotropic classical model on the cube lattice [172, 173]:

$$-\beta E = -K \sum_{\vec{j}} s_{\vec{j}} s_{\vec{j}+\vec{e}_x} s_{\vec{j}+\vec{e}_y} s_{\vec{j}+\vec{e}_x+\vec{e}_y} - J_z \sum_{\vec{j}} s_{\vec{j}} s_{\vec{j}+\vec{e}_z}. \tag{3.31}$$

---

[15]The global constraint can be seen as follows. Each configuration of $m$'s specifies a profile of operators $U$'s. Since the product of all $U$ operators is a trivial operator as shown in (3.2), and the product of all $U$ corresponds to all $m$'s being 1, we should identify $(m^y_i, m^x_j) \simeq (m^y_i + 1, m^x_j + 1)$ for all $i, j$.

where $s = \pm 1$ is put on each vertex. The first term is between four spins of all plaquettes in the $xy$ planes and the second term is between two spins over all bonds in the $z$ direction. In this classical model, the subsystem KW transformation represents a duality between the partition function of high temperature and low temperature. Numerical calculations indicate that the phase transition at the self-dual point is first order [174].

## 3.3 Subsystem KW duality defects

We switch to discussing the defects associated with the subsystem KW duality symmetry by placing one direction of the duality operator along the time. These defects are 2d surfaces with one direction along the time, and another direction along the space. For simplicity, we work on the infinite 2d square lattice. After defining the subsystem KW duality defect via the subsystem KW duality twist operator, we discuss the fusion rules as well as the mobility under translation along the space direction. For simpler conventions, we will adopt the same notation for the twist operator and the corresponding duality defect.

**Subsystem KW duality defect from subsystem KW duality twist operator:** On the infinite 2d lattice, consider a subsystem KW duality defect localized at a line $S$ that divides the space into two halves $S_+$ and $S_-$. Insertion of such a defect is realized by a twist operator $\mathcal{N}_S^{\mathrm{sub}}$ that acts on the half-space $S_-$ and terminates at $S$. The twist operator $\mathcal{N}_S^{\mathrm{sub}}$ is defined as

$$\mathcal{N}_S^{\mathrm{sub}} |\{s_{i,j}\}\rangle = \frac{1}{2^{(\ell_{S_-})/2}} \sum_{\{\widehat{s}_{i-\frac{1}{2},j-\frac{1}{2}}\}_{S_-}} (-1)^{C_S^{\mathrm{half}}} |\{\widehat{s}_{i-\frac{1}{2},j-\frac{1}{2}}\}_{S_-}; \{s_{i,j}\}_{S_+}\rangle , \tag{3.32}$$

where

$$C_S^{\mathrm{half}} := \sum_{S_-} (s_{i-1,j-1} + s_{i,j-1} + s_{i-1,j} + s_{i,j})\widehat{s}_{i-\frac{1}{2},j-\frac{1}{2}} . \tag{3.33}$$

In both equations, the summation is over the half-space $S_-$, and $\ell_{S_-}$ is a formal parameter counting the number of sites covered by $S_-$. The action of the Hermitian conjugate of the twist operator is defined as

$$(N_S^{\mathrm{sub}})^\dagger |\{\widehat{s}_{i-\frac{1}{2},j-\frac{1}{2}}\}_{S_-}; \{s_{i,j}\}_{S_+}\rangle = \frac{1}{2^{(\ell_{S_-})/2}} \sum_{\{s_{i,j}\}_{S_-}} (-1)^{C_S^{\mathrm{half}}} |\{s_{i,j}\}\rangle . \tag{3.34}$$

In Fig. 6, we give several typical examples of subsystem KW duality defects.

**Fusion rule:** We can derive the fusion rules of the subsystem KW duality defects by acting the corresponding twist operators on the same region. Here we list fusion rules of the twist operators

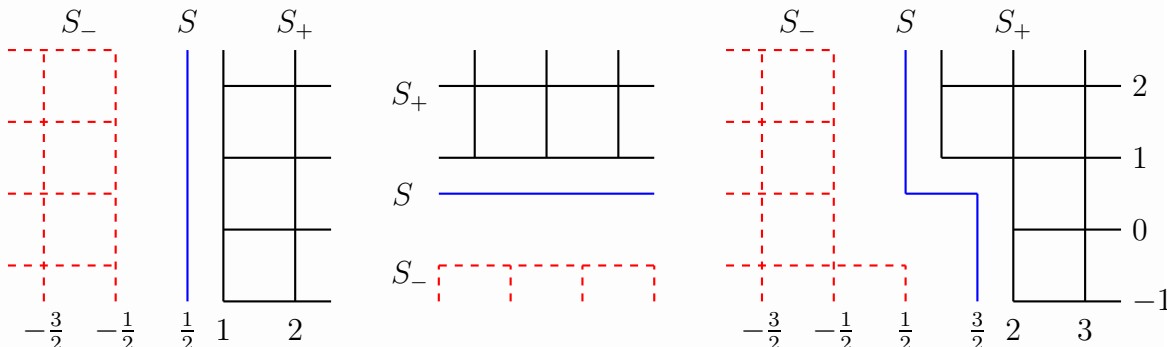

(a) Defect localized at $j \geq \frac{1}{2}$. (b) Defect localized at $j \geq \frac{1}{2}$. (c) Defect localized at $\star$ : $i = \frac{1}{2}, j \geq \frac{1}{2}$, $j = \frac{1}{2}, \frac{1}{2} \leq i \leq \frac{3}{2}$, and $i = \frac{3}{2}, j \leq \frac{1}{2}$.

Figure 6: Three examples of subsystem KW duality defects. An action of a twist operator on the half-space $S_-$ amounts to an insertion of a duality defect localized at line $S$.

shown in Fig. 6,

$$(\mathcal{N}^{\text{sub}}_{i=\frac{1}{2}})^\dagger \times \mathcal{N}^{\text{sub}}_{i=\frac{1}{2}} = \sum_{m^y_i, m^x_j = 0,1} \prod_{i \leq 0} (U^y_i)^{m^y_i} \prod_j (U^{xt}_{0,j})^{m^x_j} ,$$

$$(\mathcal{N}^{\text{sub}}_{j=\frac{1}{2}})^\dagger \times \mathcal{N}^{\text{sub}}_{j=\frac{1}{2}} = \sum_{m^y_i, m^x_j = 0,1} \prod_i (U^{yt}_{0,i})^{m^y_i} \prod_{j \leq 0} (U^x_j)^{m^x_j} , \qquad (3.35)$$

$$(\mathcal{N}^{\text{sub}}_\star)^\dagger \times \mathcal{N}^{\text{sub}}_\star = \sum_{m^y_i, m^x_j = 0,1} \prod_{i \leq 0} (U^y_i)^{m^y_i} (U^{yt}_{0,1})^{m^y_1} \prod_{j > 0} (U^{xt}_{0,j})^{m^x_j} \prod_{j \leq 0} (U^{xt}_{1,j})^{m^x_j} ,$$

where twist operators $\mathcal{N}^{\text{sub}}_{i=\frac{1}{2}}$, $\mathcal{N}^{\text{sub}}_{j=\frac{1}{2}}$ and $\mathcal{N}^{\text{sub}}_\star$ correspond to defects located at $i = \frac{1}{2}$, $j = \frac{1}{2}$ and $\star$ respectively.

Let us interpret the first fusion rule in (3.35). On the right-hand side, each term in the sum is a product of symmetry operators $U^y_i$ and twist operators $U^{xt}_{0,j}$. Note that appending a symmetry operator, e.g. $U^y_i$, to a twist operator does not change the twisted Hamiltonian (since the symmetry operator commutes with the Hamiltonian). Hence the twist operator on the right-hand side of (3.35) should be defined up to symmetry operators. Hence (3.35) simplifies to

$$(\mathcal{N}^{\text{sub}}_{i=\frac{1}{2}})^\dagger \times \mathcal{N}^{\text{sub}}_{i=\frac{1}{2}} = \sum_{m^x_j = 0,1} \prod_j (U^{xt}_{0,j})^{m^x_j} ,$$

$$(\mathcal{N}^{\text{sub}}_{j=\frac{1}{2}})^\dagger \times \mathcal{N}^{\text{sub}}_{j=\frac{1}{2}} = \sum_{m^y_i = 0,1} \prod_i (U^{yt}_{0,i})^{m^y_i} , \qquad (3.36)$$

$$(\mathcal{N}^{\text{sub}}_\star)^\dagger \times \mathcal{N}^{\text{sub}}_\star = \sum_{m^y_1, m^x_j = 0,1} (U^{yt}_{0,1})^{m^y_1} \prod_{j > 0} (U^{xt}_{0,j})^{m^x_j} \prod_{j \leq 0} (U^{xt}_{1,j})^{m^x_j} .$$

The right-hand side is again a grid defect, but the grid is only along the time direction. See Fig. 7 for the first fusion rule in (3.36). The fusion rules of subsystem KW duality defects in (3.36) are different from those of the subsystem KW duality operators in (3.25). That there are differences between space-like operators and time-like defects is a typical feature of subsystem symmetries, which are inherently non-relativistic.

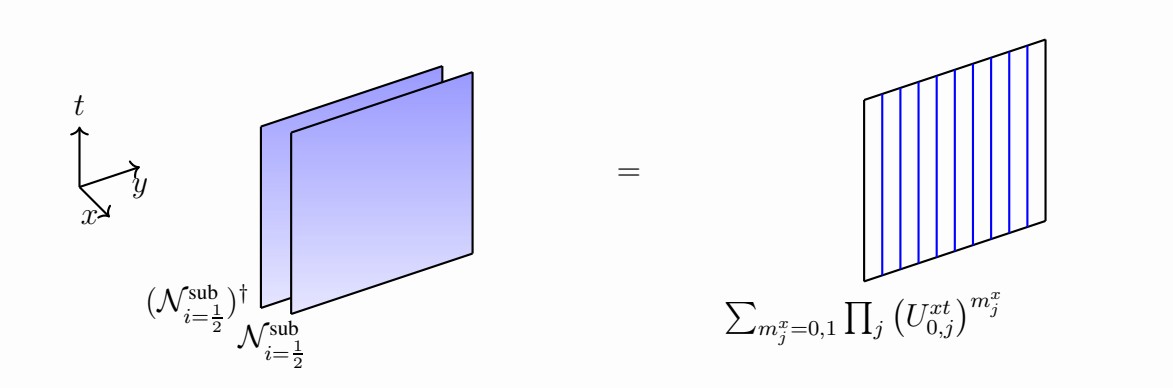

Figure 7: Fusion between two subsystem KW duality defects gives rise to a grid defect, where the grid is along the time direction.

Next, let us consider the fusion rules of the $\mathbb{Z}_2$ defects and the subsystem KW duality defects. To fuse them, the $\mathbb{Z}_2$ defect should be inserted in the surface where the duality defect resides. In other words, the corresponding $\mathbb{Z}_2$ twist operator should terminate on $S$ when subsystem KW twist operator $\mathcal{N}_S^{\mathrm{sub}}$ is considered. For simplicity, consider the fusion of $\mathcal{N}_{i=\frac{1}{2}}^{\mathrm{sub}}$, $U_{0,j}^{xt}$ and $\widehat{U}_{-\frac{1}{2},j-\frac{1}{2}}^{xt}$. The fusion rules are

$$
\begin{aligned}
\mathcal{N}_{i=\frac{1}{2}}^{\mathrm{sub}} \times U_{0,j}^{xt} &= \mathcal{N}_{i=\frac{1}{2}}^{\mathrm{sub}}, \\
\widehat{U}_{-\frac{1}{2},j-\frac{1}{2}}^{xt} \times \mathcal{N}_{i=\frac{1}{2}}^{\mathrm{sub}} &= \mathcal{N}_{i=\frac{1}{2}}^{\mathrm{sub}} \sigma_{0,j-1}^z \sigma_{0,j}^z \sim \mathcal{N}_{i=\frac{1}{2}}^{\mathrm{sub}}.
\end{aligned}
\tag{3.37}
$$

Here we identify two defects related by a local unitary operator in the same equivalent class. Therefore, we find the same fusion rules as (3.19) and (3.20).

**Mobility of subsystem KW duality defects:** We proceed to examine the mobility of the subsystem KW duality defects. Intuitively, since the subsystem symmetry is not mobile under a generic translation along the space, so does the subsystem KW duality defect. However, below we will find that this is not true. We will demonstrate the mobility by studying the example in Fig. 8, where the deformation happens only around the origin.

We find a local unitary operator $W$ relating the two duality twist operators

$$
\mathcal{N}_{S'}^{\mathrm{sub}} = \mathcal{N}_S^{\mathrm{sub}} W,
\tag{3.38}
$$

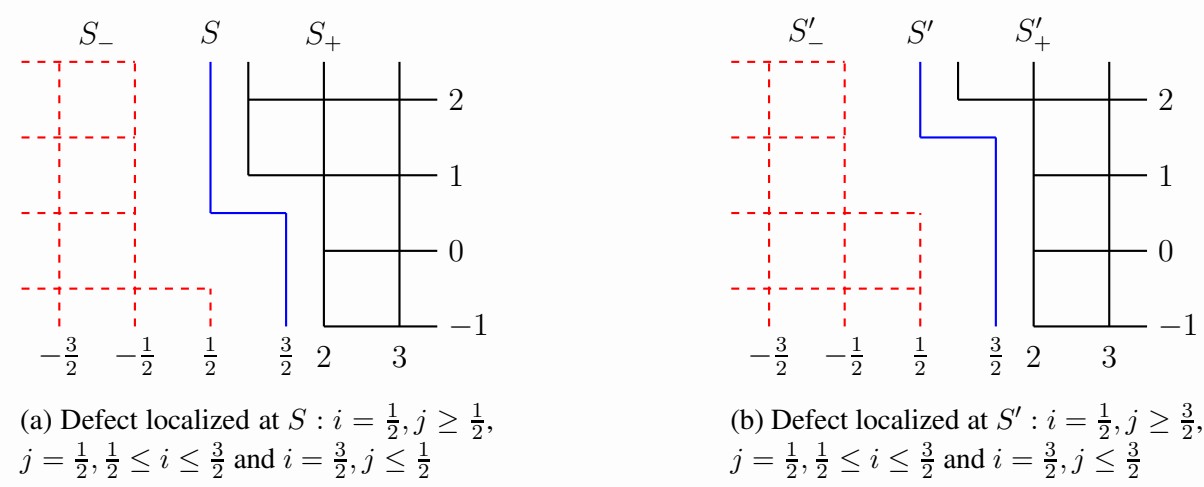

(a) Defect localized at $S : i = \frac{1}{2}, j \geq \frac{1}{2}$, $j = \frac{1}{2}, \frac{1}{2} \leq i \leq \frac{3}{2}$ and $i = \frac{3}{2}, j \leq \frac{1}{2}$

(b) Defect localized at $S' : i = \frac{1}{2}, j \geq \frac{3}{2}$, $j = \frac{1}{2}, \frac{1}{2} \leq i \leq \frac{3}{2}$ and $i = \frac{3}{2}, j \leq \frac{3}{2}$

Figure 8: Duality defects located on $S$ and $S'$. We show that one can be moved to the other by a local unitary operator.

where

$$W = \mathsf{CZ}_{(0,0),(1,1)}\mathsf{CZ}_{(1,0),(1,1)}\mathsf{CZ}_{(0,1),(1,1)}\mathsf{H}_{(1,1)} . \tag{3.39}$$

We adopt the same convention as in the previous section, i.e. $\mathsf{CZ}_{(i,j),(i',j')}$ is the control Z gate acting on sites $(i,j)$ and $(i',j')$, and $\mathsf{H}_{(1,1)}$ is the Hadamard gate acting on site $(1,1)$. (3.38) can be checked directly

$$\mathcal{N}_S^{\text{sub}}W\,|\{s_{i,j}\}\rangle = \mathcal{N}_S^{\text{sub}}\frac{1}{\sqrt{2}}\sum_{s'_{1,1}}(-1)^{(s_{0,0}+s_{0,1}+s_{1,0}+s_{1,1})s'_{1,1}}\,|\{s_{i,j}\}_{S_-};s'_{1,1};\{s_{i,j}\}_{S'_+}\rangle$$

$$= \frac{1}{2^{(\ell_{S_-}+1)/2}}\sum_{\{\widehat{s}_{i-\frac{1}{2},j-\frac{1}{2}}\}_{S_-},s'_{1,1}}(-1)^{C_S^{\text{half}}+(s_{0,0}+s_{0,1}+s_{1,0}+s_{1,1})s'_{1,1}}\,|\{\widehat{s}_{i-\frac{1}{2},j-\frac{1}{2}}\}_{S_-};s'_{1,1};\{s_{i,j}\}_{S'_+}\rangle$$

$$\overset{s'_{1,1}\to\widehat{s}_{\frac{1}{2},\frac{1}{2}}}{=}\frac{1}{2^{(\ell_{S'_-})/2}}\sum_{\{\widehat{s}_{i-\frac{1}{2},j-\frac{1}{2}}\}_{S'_-}}(-1)^{C_{S'}^{\text{half}}}\,|\{\widehat{s}_{i-\frac{1}{2},j-\frac{1}{2}}\}_{S'_-};\{s_{i,j}\}_{S'_+}\rangle = \mathcal{N}_{S'}^{\text{sub}}\,|\{s_{i,j}\}\rangle .$$

$$\tag{3.40}$$

This shows that the two defects $\mathcal{N}_{S'}^{\text{sub}}$ and $\mathcal{N}_S^{\text{sub}}$ are equivalent, hence the defect is mobile under a spatial deformation. As commented at the end of Sec. 2.3, this does not prove that the subsystem KW generator is topological under arbitrary spacetime deformation. We will not try to prove this in this work.

## 3.4 Subsystem KW duality symmetry

We briefly comment on the subsystem KW transformation with a single Hilbert space. Similar to the $(1+1)$d transformation, we will define a new transformation $\bar{\mathcal{N}}^{\text{sub}}$ by redefining the plaquette

spins to sites,

$$\widehat{s}_{i+\frac{1}{2},j+\frac{1}{2}} \to s'_{i,j} \tag{3.41}$$

Concretely, (3.14) becomes

$$\bar{\mathcal{N}}^{\text{sub}} \left| \{s_{i,j}\} \right\rangle = \frac{1}{2^{(L_x+L_y)/2}} \sum_{\{s'_{i,j}\}} (-1)^{C_{\text{bulk}}+C_{\text{bdy}}} \left| \{s'_{i,j}\} \right\rangle , \tag{3.42}$$

where the bulk and boundary terms are

$$C_{\text{bulk}} := \sum_{i=1}^{L_x} \sum_{j=1}^{L_y} (s_{i,j} + s_{i,j+1} + s_{i+1,j} + s_{i+1,j+1}) s'_{i,j} ,$$

$$C_{\text{bdy}} := \sum_{j=1}^{L_y} t'^x_j s_{L_x+1,j} + \sum_{i=1}^{L_x} t'^y_i s_{i,L_y+1} + t'^{xy} s_{L_x+1,L_y+1} , \tag{3.43}$$

$\bar{\mathcal{N}}^{\text{sub}}$ is a symmetry of the critical plaquette Ising model because it exchanges the plaquette-$\sigma^z$ terms and the transverse-$\sigma^x$ term

$$\bar{\mathcal{N}}^{\text{sub}} \sigma^z_{i,j} \sigma^z_{i+1,j} \sigma^z_{i,j+1} \sigma^z_{i+1,j+1} \left| \psi \right\rangle = \sigma^x_{i,j} \bar{\mathcal{N}}^{\text{sub}} \left| \psi \right\rangle , \quad \forall \left| \psi \right\rangle \in \mathcal{H} ,$$

$$\bar{\mathcal{N}}^{\text{sub}} \sigma^x_{i,j} \left| \psi \right\rangle = \sigma^z_{i,j} \sigma^z_{i-1,j} \sigma^z_{i,j-1} \sigma^z_{i-1,j-1} \bar{\mathcal{N}}^{\text{sub}} \left| \psi \right\rangle , \quad \forall \left| \psi \right\rangle \in \mathcal{H} . \tag{3.44}$$

In the untwisted sector, the nontrivial fusion rules is

$$\bar{\mathcal{N}}^{\text{sub}} \times \bar{\mathcal{N}}^{\text{sub}} = \prod_{i=1}^{L_x} \left(1 + U^y_i\right) \prod_{j=1}^{L_y} \left(1 + U^x_j\right) T_{xy} , \tag{3.45}$$

where $T_{xy}$ implements the lattice translation in the diagonal direction

$$T_{xy} \left| \{s_{i,j}\} \right\rangle = \left| \{s'_{i,j} = s_{i+1,j+1}\} \right\rangle \tag{3.46}$$

In a similar way, the symmetry defects are created by acting the twist operators

$$\bar{\mathcal{N}}^{\text{sub}}_S \left| \{s_{i,j}\} \right\rangle = \frac{1}{2^{(\ell_{S_-})/2}} \sum_{\{s'_{i,j}\}_{S_-}} (-1)^{\sum_{S_-} (s_{i,j}+s_{i+1,j}+s_{i,j+1}+s_{i+1,j+1}) s'_{i,j}} \left| \{s'_{i,j}\}_{S_-}; \{s_{i,j}\}_{S_+} \right\rangle \tag{3.47}$$

defined on the half space. The fusion rules and mobility of these defects follows in the similar discussion as Sec. 3.3.

## 3.5 Anomaly of subsystem KW duality symmetry

We conclude this section by generalizing the approach in Sec. 2.5 to show that the non-invertible subsystem KW duality symmetry is anomalous.

We proceed to prove the anomaly by contradiction. Assume that the subsystem $\mathbb{Z}_2$ symmetry is anomaly free, hence it is compatible with a gapped phase with one ground state, i.e. subsystem

symmetry protected topological phase. Such a phase has been classified in [159], which is given by $H^2(\mathbb{Z}_2 \times \mathbb{Z}_2, U(1))/(H^2(\mathbb{Z}_2, U(1)))^3 = \mathbb{Z}_2$.[16] Hence there is one trivial phase and one non-trivial phase. The trivial phase has partition function $Z_{\text{triv}} = 1$, and the non-trivial phase has the partition function [161]

$$Z_{\text{SSPT}}[A_\tau, A_{xy}] = \int \mathcal{D}\phi^{xy}(-1)^{\int \phi^{xy}(\partial_\tau A_{xy} - \partial_x \partial_y A_\tau) + A_\tau A_{xy}}, \tag{3.48}$$

where $\phi^{xy} \in 0, 1$ is an auxiliary field, integrating which constraints the flatness condition of the background field $A_\tau, A_{xy}$ of the subsystem $\mathbb{Z}_2$ gauge field. See Appendix A or [89] for further discussions on the relation between the subsystem $\mathbb{Z}_2$ symmetry operators (3.1) and the subsystem $\mathbb{Z}_2$ symmetry background fields on the square lattice. The Lagrangian is invariant under gauge transformation $\phi^{xy} \to \phi^{xy} - \alpha, A_\tau \to A_\tau + \partial_\tau \alpha, A_{xy} \to A_{xy} + \partial_x \partial_y \alpha$.

Let us check whether $Z_{\text{triv}}$ and $Z_{\text{SSPT}}$ are separately invariant under the subsystem KW transformation, i.e. the gauging of the subsystem $\mathbb{Z}_2$ symmetry. For $Z_{\text{triv}}$, gauging subsystem $\mathbb{Z}_2$ symmetry gives a spontaneous subsystem symmetry broken (SSSB) phase, whose partition function is [17]

$$\begin{aligned} Z_{\text{SSSB}} &= \int \mathcal{D}a \, (-1)^{\int a_\tau A_{xy} + a_{xy} A_\tau} \\ &= \sum_{\text{conf. of } a} (-1)^{\sum_{i=1}^{L_x}(W^a_{\tau,y;i} W^A_{y;i} + W^A_{\tau,y;i+\frac{1}{2}} W^a_{y;i+\frac{1}{2}}) + \sum_{j=1}^{L_y}(W^a_{\tau,x;j} W^A_{x;j} + W^A_{\tau,x;j+\frac{1}{2}} W^a_{x;j+\frac{1}{2}})}, \end{aligned} \tag{3.49}$$

where we regulate the integration of the dynamical gauge field on a finite cubic lattice with $L_x \times L_y$ sites and suppressed the overall normalization. The sum runs over all configurations of gauge field $a$, which can be converted to the summation over holonomy variables. However, there are only $2(L_x + L_y - 1)$ independent holonomy variables out of $2(L_x + L_y)$ variables

$$W^a_{\tau,y;i}, W^a_{y;i+\frac{1}{2}}, W^a_{\tau,x;j}, W^a_{x;j+\frac{1}{2}} \in \{0, 1\}. \tag{3.50}$$

To pick the independent variables, recall the constraint on the space holonomy variables

$$\sum_{i=1}^{L_x} W^a_{y;i+\frac{1}{2}} + \sum_{j=1}^{L_y} W^a_{x;j+\frac{1}{2}} = \sum_{i=1}^{L_x} W^A_{y;i} + \sum_{j=1}^{L_y} W^A_{x;j} = 0, \tag{3.51}$$

which also shows that the partition function is invariant under a global unit shift of the time holon-

---

[16]The classification is only for strong SSPT phases, while there are also weak SSPT phases. The weak SSPT phases only made use of a subset of symmetry generators, say $\{U^x_j\}$ but not $\{U^y_i\}$, hence the background fields are different. In [161], the weak SSPT with subsystem $\mathbb{Z}_2$ symmetry was discussed, whose field theory is $A_t \partial_x A_y$. Below, we would like to discuss the anomaly of the *entire* subsystem $\mathbb{Z}_2$ symmetry whose background fields are $A_t, A_{xy}$, thus we will only discuss the SSPT whose field theory is in terms of $A_t, A_{xy}$. It turns out that the only SSPT with such a background field is the strong SSPT.

[17]Again we will suppress the normalization factor.

omy variables. With the above redundancy, we can choose the following gauge fixing condition

$$W^a_{\tau,y;1} = W^A_{\tau,y;\frac{3}{2}} = 0 \,,$$

$$W^a_{y;\frac{3}{2}} = \sum_{i=2}^{L_x} W^a_{y;i+\frac{1}{2}} + \sum_{j=1}^{L_y} W^a_{x;j+\frac{1}{2}} \,,$$

$$W^A_{y;1} = \sum_{i=2}^{L_x} W^A_{y;i} + \sum_{j=1}^{L_y} W^A_{x;j} \,.$$

(3.52)

Then we can evaluate the partition function (3.49)

$$Z_{\text{SSSB}} = \sum_{\text{conf. of } a} (-1)^{\sum_{i=2}^{L_x}(W^a_{\tau,y;i}W^A_{y;i}+W^A_{\tau,y;i+\frac{1}{2}}W^a_{y;i+\frac{1}{2}})+\sum_{j=1}^{L_y}(W^a_{\tau,x;j}W^A_{x;j}+W^A_{\tau,x;j+\frac{1}{2}}W^a_{x;j+\frac{1}{2}})}$$

$$= \prod_{i=2}^{L_x} \delta(W^A_{\tau,y;i+\frac{1}{2}})\delta(W^A_{y;i}) \prod_{j=1}^{L_y} \delta(W^A_{\tau,x;j+\frac{1}{2}})\delta(W^A_{x;j}) \,.$$

(3.53)

Indeed, the ground state degeneracy is extensive and spontaneously breaks the subsystem $\mathbb{Z}_2$ symmetry. This means that the trivial phase is not invariant under gauging.

For $Z_{\text{SSPT}}$, gauging subsystem $\mathbb{Z}_2$ symmetry yields a partially spontaneous subsystem symmetry broken (PSSSB) phase, whose partition function is

$$Z_{\text{PSSSB}} = \int \mathcal{D}a \int \mathcal{D}\phi^{xy} (-1)^{\int \phi^{xy}(\partial_\tau a_{xy} - \partial_x\partial_y a_\tau) + a_\tau a_{xy} + a_\tau A_{xy} + a_{xy} A_\tau}$$

$$= \int \mathcal{D}a (-1)^{\int a_\tau a_{xy} + a_\tau A_{xy} + a_{xy} A_\tau} \,,$$

(3.54)

where we first integrated out the auxiliary field $\phi^{xy}$ enforcing the flatness condition $\partial_\tau a_{xy} - \partial_x\partial_y a_\tau = 0$. PSSSB means the ground state degeneracy is not extensive and only part of the subsystem symmetry is broken.

To see this, let us integrate out $a_\tau, a_{xy}$ in (3.54). Naively, one would attempt to simply integrate out $a_\tau$ which enforces $a_{xy} = A_{xy}$. Such a naive integration, however, is obstructed due to subtle global constraints originating from the flatness condition for the gauge field. Below, we evaluate the integration carefully by first converting the integration of gauge fields in terms of summation over holonomy variables. Concretely,

$$Z_{\text{PSSSB}} = \sum_{\{\text{conf. of } a\}} (-1)^{\sum_{i=1}^{L_x} W^a_{y;i+\frac{1}{2}}(W^a_{\tau;y,i}+W^a_{\tau,y;i+1})+\sum_{j=1}^{L_y} W^a_{x;j+\frac{1}{2}}(W^a_{\tau,x;j}+W^a_{\tau,x;j+1})}$$

$$\times (-1)^{\sum_{i=1}^{L_x}(W^a_{\tau,y;i}W^A_{y;i}+W^A_{\tau,y;i+\frac{1}{2}}W^a_{y;i+\frac{1}{2}})+\sum_{j=1}^{L_y}(W^a_{\tau,x;j}W^A_{x;j}+W^A_{\tau,x;j+\frac{1}{2}}W^a_{x;j+\frac{1}{2}})} \,.$$

(3.55)

Using the gauge fixing condition (3.52) and summing over the spatial holonomy variables of the dynamical gauge field $W^a_{y;i+\frac{1}{2}}, W^a_{x;j+\frac{1}{2}}$ for $i = 2, ..., L_x$ and $j = 1, ..., L_y$, we will get

$$Z_{\text{PSSSB}} = \sum_{W^a_{\tau,y;i+\frac{1}{2}},W^a_{\tau,x;j+\frac{1}{2}}=0,1} \tilde{\delta}_{A,a} (-1)^{\sum_{i=2}^{L_x} W^a_{\tau,y;i}W^A_{y;i}+\sum_{j=1}^{L_y} W^a_{\tau,x;j}W^A_{x;j}} \,,$$

(3.56)

where the delta constraint $\tilde{\delta}_{A,a}$ includes

$$W^A_{\tau,y;i+\frac{1}{2}} = W^a_{\tau,y;2} + W^a_{\tau,y;i} + W^a_{\tau,y;i+1}, \quad i = 2, ..., L_x - 1$$

$$W^A_{\tau,y;L_x+\frac{1}{2}} = W^a_{\tau,y;2} + W^a_{\tau,y;L_x}, \tag{3.57}$$

$$W^A_{\tau,x;j+\frac{1}{2}} = W^a_{\tau,y;2} + W^a_{\tau,x;j} + W^a_{\tau,x;j+1}, \quad j = 1, ..., L_y$$

The first two lines are constraints on the $y$ direction and the last line is on the $x$ direction. Further summing over the delta constraints on $y$ and $x$ directions separately leads to

$$\sum_{i=2}^{L_x} W^A_{\tau,y;i+\frac{1}{2}} = L_x W^a_{\tau,y;2},$$

$$\sum_{j=1}^{L_y} W^A_{\tau,x;j+\frac{1}{2}} = L_y W^a_{\tau,y;2}. \tag{3.58}$$

This means that the background field $A$ obey global constraints, depending on the parity of $L_x$ and $L_y$:

1. $(L_x, L_y) = (\text{even}, \text{even})$: two independent constraints $\sum_{i=2}^{L_x} W^A_{\tau,y;i+\frac{1}{2}} = \sum_{j=1}^{L_y} W^A_{\tau,x;j+\frac{1}{2}} = 0$, leading to four-fold ground state degeneracy.

2. $(L_x, L_y) = (\text{even}, \text{odd})$: one independent constraint $\sum_{i=2}^{L_x} W^A_{\tau,y;i+\frac{1}{2}} = 0$, leading to two-fold ground state degeneracy.

3. $(L_x, L_y) = (\text{odd}, \text{even})$: one independent constraint $\sum_{j=1}^{L_y} W^A_{\tau,x;j+\frac{1}{2}} = 0$, leading to two-fold ground state degeneracy.

4. $(L_x, L_y) = (\text{odd}, \text{odd})$: one independent constraint $\sum_{i=2}^{L_x} W^A_{\tau,y;i+\frac{1}{2}} + \sum_{j=1}^{L_y} W^A_{\tau,x;j+\frac{1}{2}} = 0$, leading to two-fold ground state degeneracy.

For all cases, the theory after subsystem KW transformation spontaneously breaks part of the symmetry, and therefore $Z_{\text{SSPT}}$ is not invariant under gauging subsystem $\mathbb{Z}_2$ symmetry. [18]

Because both $Z_{\text{triv}}$ and $Z_{\text{SSPT}}$ are not compatible with subsystem KW duality symmetry, we can conclude that this symmetry is anomalous. We remark that this anomaly protects the phase transition of the plaquette Ising model with the critical transverse field in (3.29).

# 4 Discussion and future directions

In this paper, we gave the first example of a subsystem non-invertible symmetry — the subsystem KW duality symmetry — in lattice models in $(2 + 1)$d, thereby filling in one missing corner of

---

[18]The feature where the gauged strong SSPT has 4 or 2 GSD depending on the parity of $(L_x, L_y)$ can also be seen for the lattice model $H_{\text{SSPT}} = -\sum_{i,j} \sigma^x_{i,j} \sigma^z_{i,j+1} \sigma^z_{i+1,j} \sigma^z_{i+1,j+1} \sigma^z_{i,j-1} \sigma^z_{i-1,j} \sigma^z_{i-1,j-1}$. We thank Trithep Devakul for pointing out this exactly solvable model.

| | duality symmetry | subsystem duality symmetry |
|---|---|---|
| dimension | $(1+1)$d | $(2+1)$d |
| operation | gauging $\mathbb{Z}_2$ | gauging subsystem $\mathbb{Z}_2$ |
| fusion rules of operators | (2.6) | (3.19), (3.20), (3.28) |
| fusion rules of defects | (2.15) | (3.36) |
| invertible defect mobility | mobile | restricted mobile |
| duality defect mobility | mobile | mobile along spatial translation |
| anomaly of duality symmetry | anomalous | anomalous |

Table 2: Comparison between the KW duality symmetry in $(1 + 1)$d and the subsystem KW duality symmetry in $(2 + 1)$d.

generalized symmetries in Tab. 1. The discussion of subsystem duality symmetry in $(2 + 1)$d is a direct generalization of the ordinary duality symmetry in $(1 + 1)$d. We listed the comparison in Tab. 2. Here we comment on several future directions.

To have a generic understanding of subsystem non-invertible symmetry, one can employ similar analysis to models with general abelian subsystem symmetry [83,175] or theories with global dipole symmetry [87,91]. Another interesting direction is to find the last missing corner in Tab. 1, a higher subsystem non-invertible symmetry, by studying duality operators and defects from gauging higher-form subsystem symmetry [85,90,142–144,146,148,176–178].

As a duality transformation, the subsystem KW transformation provides a new method to study SSPT phases. For example, we can perform a parallel analysis for $(2 + 1)$d many-body systems with $\mathbb{Z}_2 \times \mathbb{Z}_2$ subsystem symmetries where subsystem KW transformation $\mathcal{N}^{\text{sub}}$ amounts to gauging the $\mathbb{Z}_2 \times \mathbb{Z}_2$ subsystem symmetries. Moreover, there exists a unitary decorated domain wall (DW) transformation $U_{\text{DW}}$ relating $\mathbb{Z}_2 \times \mathbb{Z}_2$ strong SSPT phase and trivially gapped phase [19] in $(2+1)$d [79,179]. Similar to the cases in $(1+1)$d [154,180,181], one can define the subsystem Kennedy-Tasaki (KT) transformation

$$\mathcal{N}_{\text{KT}} = \mathcal{N}^{\text{sub}} U_{\text{DW}} \mathcal{N}^{\text{sub}}, \tag{4.1}$$

which relates the SSPT to SSSB phase, while leaving the trivial gapped phase invariant. See Fig. 9 for a summary. The KT duality transformation thus offers a hidden symmetry-breaking interpretation for strong SSPT phase [182–184]. It is interesting to explore the explicit expression of this duality transformation and its applications to gapped and gapless SSPTs [181,185–188].

In [89], we discussed the subsystem Jordan-Wigner (JW) transformation. It is known [169, 189,190] that in $(1+1)$d, the JW transformation, the KW transformation, and stacking a fermionic

---

[19]Here we consider the product state, which excludes the weak SSPT phase.

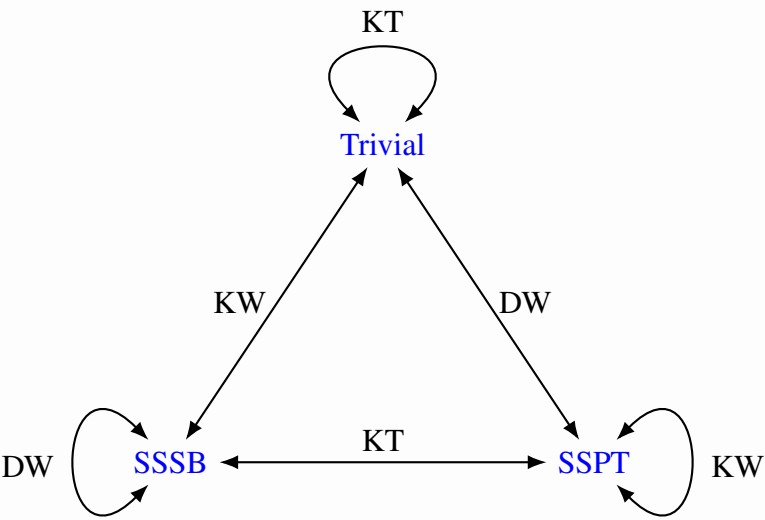

Figure 9: Three gapped phases with $\mathbb{Z}_2 \times \mathbb{Z}_2$ subsystem symmetry and the dualities between them. Note that the KW duality transformation in the figure applies to the entire subsystem $\mathbb{Z}_2 \times \mathbb{Z}_2$ symmetry.

SPT, i.e. the Arf invariant, satisfy the following relation

$$(\text{Stack fermionic SPT}) \cdot \text{JW} = \text{JW} \cdot \text{KW}. \tag{4.2}$$

In $(2+1)$d systems with subsystem $\mathbb{Z}_2$ symmetries, besides the subsystem JW transformation, subsystem KW transformation, and stacking a subsystem fermionic SPT (i.e. the subsystem Arf invariant in [89, Sec. 4]), we can also stack a bosonic subsystem $\mathbb{Z}_2$ strong and weak SPT [158, 161, 191, 192], see also the discussion in Sec. 3.5. Hence these four operators generate a more complicated diagram. It would be interesting to explore the relation between them in the future.

Finally, it has been realized recently [164, 193] that gaugings and measurements can be implemented in the quantum circuits to prepare interesting topological states, and has potential application in fault-tolerant quantum computation. It is also widely appreciated [194–199] that quantum codes with subsystem symmetries behave better in protecting quantum information from errors. It would be interesting to investigate whether the subsystem KW duality operators can be implemented in the quantum circuits and to search for possible applications in quantum computation.

# Acknowledgements

W.C. thanks Kantaro Ohmori for his wonderful lectures on non-invertible symmetry in the Kavli Winter Asian School in IBS, Korea and his insights during the discussion. We also thank Trithep Devakul, Jie Wang, Satoshi Yamaguchi, Shi Chen, and Han Yan for helpful discussions, and

Masaki Oshikawa for the collaboration on a related work [154] and discussions. This work is partially supported by World Premier International Research Center Initiative (WPI) Initiative, MEXT, Japan at Kavli IPMU, the University of Tokyo. W.C. and L.L. are supported by the Global Science Graduate Course (GSGC) program of the University of Tokyo. W.C. also acknowledges support from JSPS KAKENHI grant numbers JP19H05810, JP22J21553 and JP22KJ1072. M.Y. is also supported in part by the JSPS Grant-in-Aid for Scientific Research (19K03820, 19H00689, 20H05860, 23H01168), and by JST, Japan (PRESTO Grant No. JPMJPR225A, Moonshot R&D Grant No. JPMJMS2061). The authors of this paper were ordered alphabetically.

# A  Gauging subsystem $\mathbb{Z}_2$ symmetry

The subsystem KW transformation gauges a subsystem $\mathbb{Z}_2$ symmetry. We will review the technical details [89] in this appendix.

**Turing on background gauge fields:**  To couple the theory to a background gauge field, consider a spacetime cubic lattice with $T$ sites along the time direction and $L_x \times L_y$ sites along the space direction. The background gauge fields $B^{\tau}_{i,j,k+\frac{1}{2}}, B^{xy}_{i+\frac{1}{2},j+\frac{1}{2},k} \in \{0,1\}$ are defined on the time link and the spatial plaquette respectively, see Fig.10.

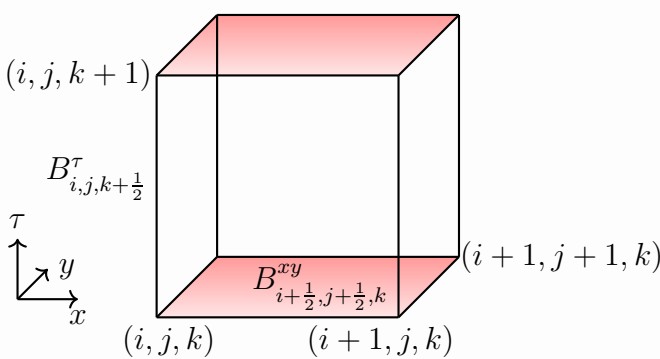

Figure 10: Background gauge field for subsystem $\mathbb{Z}_2$ symmetry on lattice.

The gauge invariant holonomy variables $W^B_{\tau;i,j}, W^B_{x;j+\frac{1}{2}}, W^B_{y;i+\frac{1}{2}} \in \{0,1\}$ are defined by sum-

ming gauge fields over particular spacetime circles.

$$W^B_{\tau;i,j} = \sum_{k=1}^{T} B^{\tau}_{i,j,k+\frac{1}{2}} = W^B_{\tau,x,j} + W^B_{\tau,y,i}\,,$$

$$W^B_{x;j+\frac{1}{2}} = \sum_{i=1}^{L_x} B^{xy}_{i+\frac{1}{2},j+\frac{1}{2},k} = \mathfrak{t}^x_{j+\frac{1}{2}}\,, \tag{A.1}$$

$$W^B_{y;i+\frac{1}{2}} = \sum_{j=1}^{L_y} B^{xy}_{i+\frac{1}{2},j+\frac{1}{2},k} = \mathfrak{t}^y_{i+\frac{1}{2}}\,.$$

The number of holonomy variables grows with system size. Because the $L_x \times L_y$ variables $W^B_{\tau,i,j}$ are highly reducible, instead we use $W^B_{\tau,x;j}, W^B_{\tau,y;i} \in \{0,1\}$, detecting the insertion of symmetry operator $(U^x_j)^{W^B_{\tau,x;j}}$ and $(U^y_i)^{W^B_{\tau,y;i}}$. The constraint on the symmetry generator $\prod_{i=1}^{L_x} U^y_i \prod_{j=1}^{L_y} U^x_j = 1$ induces the redundancy

$$W^B_{\tau,x;j} \to W^B_{\tau,x;j} + 1\,, \quad W^B_{\tau,y;i} \to W^B_{\tau,y;i} + 1\,, \tag{A.2}$$

which is obvious from (A.1). Along the spatial cycle, the holonomy variables $W^B_{x;j+\frac{1}{2}}, W^B_{y;i+\frac{1}{2}}$ detect the boundary conditions with the following constraint

$$\sum_{i=1}^{L_x} W^B_{y;i+\frac{1}{2}} = \sum_{j=1}^{L_y} W^B_{x;j+\frac{1}{2}} = t^{xy}\,. \tag{A.3}$$

In summary, there are $L_x + L_y - 1$ independent holonomy variables along the time and space directions separately. The partition function with background fields is

$$Z(W^B_{\tau,x;j}, W^B_{\tau,y;i}, W^B_{x;j+\frac{1}{2}}, W^B_{y;i+\frac{1}{2}}) := \mathrm{Tr}_{\left\{ W^B_{x;j+\frac{1}{2}}, W^B_{y;i+\frac{1}{2}} \right\}} \left( \prod_{j=1}^{L_y} (U^x_j)^{W^B_{t,x;j}} \right) \left( \prod_{i=1}^{L_x} (U^y_i)^{W^B_{t,y;i}} \right) e^{-\beta H}\,. \tag{A.4}$$

**Gauging subsystem $\mathbb{Z}_2$ symmetry:** Consider a theory $X$ with a non-anomalous subsystem $\mathbb{Z}_2$ symmetry on the spacetime 3-torus $T^3$. After gauging, we obtain a theory $X/\mathbb{Z}_2^{\mathrm{sub}}$ living on the dual lattice with a new subsystem $\mathbb{Z}_2$ symmetry. According to the new subsystem $\mathbb{Z}_2$ symmetry, the Hilbert space is divided into symmetry and twist sectors labeled by $\widehat{\mathfrak{u}} := \{\widehat{u}^x_{j+\frac{1}{2}}, \widehat{u}^y_{i+\frac{1}{2}}\}, \widehat{\mathfrak{t}} := \{\widehat{\mathfrak{t}}^x_j, \widehat{\mathfrak{t}}^y_i\}$. Equivalently, we can couple the new theory with a new background subsystem $\mathbb{Z}_2$ gauge field.

The gauging procedure contains two steps: first attach to the partition function a phase whose exponent is the cup product of the gauge fields in $X$ and $X/\mathbb{Z}_2^{\mathrm{sub}}$ and then sum over all distinct gauge field configurations in the theory $X$ to promote its gauge field into a dynamical field. In terms of gauging ordinary $\mathbb{Z}_2$ symmetry in $(1+1)$d, the background for the quantum symmetry couples to the dynamical gauge field via the standard coupling

$$\int b_\tau B_x + b_x B_\tau = W^B_\tau W^b_x + W^B_x W^b_\tau\,. \tag{A.5}$$

(A.5) can be derived using the flatness of gauge fields $B, b$. We show it on lattice for later conve-nience of the generalization in subsystem symmetry. After discretization on the spacetime lattice with $L_x \times T$ sites, the flatness means that the gauge field $B$ is closed

$$B^\tau_{i,j+\frac{1}{2}} + B^\tau_{i+1,j+\frac{1}{2}} + B^x_{i+\frac{1}{2},j} + B^x_{i+\frac{1}{2},j+1} = 0 \,, \tag{A.6}$$

equivalent to the statement that the gauge field $B$ is exact

$$B^x_{i+\frac{1}{2},j} = B_{i,j} + B_{i+1,j}, \quad B^\tau_{i,j+\frac{1}{2}} = B_{i,j} + B_{i,j+1} \,, \tag{A.7}$$

where $B_{i,j} \in \{0,1\}$ is the potential field of the gauge field. By definition, the holonomy variables measure the twist boundary of the potential field

$$\begin{aligned}
W^B_\tau &= \sum_{j=1}^{T} B^\tau_{i,j+\frac{1}{2}} = B_{i,1} + B_{i,T+1} \,, \\
W^B_x &= \sum_{i=1}^{L_x} B^x_{i+\frac{1}{2},j} = B_{1,j} + B_{L_x+1,j} \,.
\end{aligned} \tag{A.8}$$

(A.5) then follows

$$\begin{aligned}
\int B_\tau b_x + B_x b_\tau &= \sum_{i=1}^{L_x} \sum_{j=1}^{T} B^\tau_{i,j+\frac{1}{2}} b^x_{i,j+\frac{1}{2}} + B^x_{i+\frac{1}{2},j} b^\tau_{i+\frac{1}{2},j} \\
&= \sum_{i=1}^{L_x} \sum_{j=1}^{T} (B_{i,j} + B_{i,j+1}) b^x_{i,j+\frac{1}{2}} + (B_{i,j} + B_{i+1,j}) b^\tau_{i+\frac{1}{2},j} \\
&= \sum_{i=1}^{L_x} \sum_{j=1}^{T} B_{i,j} (b^x_{i,j-\frac{1}{2}} + b^x_{i,j+\frac{1}{2}} + b^\tau_{i-\frac{1}{2},j} + b^\tau_{i+\frac{1}{2},j}) \\
&\quad + \sum_{i=1}^{L_x} b^x_{i,T+\frac{1}{2}} (B_{i,1} + B_{i,L_x+1}) + \sum_{j=1}^{T} b^\tau_{L_x+\frac{1}{2},j} (B_{1,j} + B_{T+1,j}) \\
&= W^B_\tau W^b_x + W^B_x W^b_\tau \,.
\end{aligned} \tag{A.9}$$

Similarly, we can regularize the coupling to subsystem symmetry gauge fields. Consider the spacetime cub lattice with $L_x \times L_y \times T$ sites. The flatness means that the gauge field $B$ is closed

$$B^\tau_{i,j,k+\frac{1}{2}} + B^\tau_{i+1,j,k+\frac{1}{2}} + B^\tau_{i,j+1,k+\frac{1}{2}} + B^\tau_{i+1,j+1,k+\frac{1}{2}} + B^{xy}_{i+\frac{1}{2},j+\frac{1}{2},k} + B^{xy}_{i+\frac{1}{2},j+\frac{1}{2},k+1} = 0 \,, \quad \text{(A.10)}$$

equivalent to the exactness

$$\begin{aligned}
B^\tau_{i,j,k+\frac{1}{2}} &= B_{i,j,k} + B_{i,j,k+1} \\
B^{xy}_{i+\frac{1}{2},j+\frac{1}{2},k} &= B_{i,j,k} + B_{i+1,j,k} + B_{i,j+1,k} + B_{i+1,j+1,k} \,,
\end{aligned} \tag{A.11}$$

where $B_{i,j,k} \in \{0,1\}$ is the potential field for the subsystem gauge field. The holonomy variables measure the twist boundary of the potential field

$$W_{\tau,x,j}^B + W_{\tau,y,i}^B = W_{\tau,i,j}^B = \sum_{k=1}^{T} B_{i,j,k+\frac{1}{2}}^{\tau} = B_{i,j,1} + B_{i,j,T+1}$$

$$W_{x,j+\frac{1}{2}}^B = \sum_{i=1}^{L_x} B_{i+\frac{1}{2},j+\frac{1}{2},k}^{xy} = B_{1,j,k} + B_{L_x+1,j,k} + B_{1,j+1,k} + B_{L_x+1,j+1,k}$$

$$W_{y,i+\frac{1}{2}}^B = \sum_{j=1}^{L_y} B_{i+\frac{1}{2},j+\frac{1}{2},k}^{xy} = B_{i,1,k} + B_{i,L_y+1,k} + B_{i+1,1,k} + B_{i+1,L_y+1,k}$$

(A.12)

The standard regularization of the coupling to subsystem gauge field is

$$\int B_\tau b_{xy} + B_{xy} b_\tau = \sum_{i=1}^{L_x} \sum_{j=1}^{L_y} \sum_{k=1}^{T} B_{i+\frac{1}{2},j+\frac{1}{2},k}^{xy} b_{i+\frac{1}{2},j+\frac{1}{2},k}^{\tau} + B_{i,j,k+\frac{1}{2}}^{\tau} b_{i,j,k+\frac{1}{2}}^{xy} .$$

(A.13)

After a lengthy but straightforward derivation, we obtain

$$\int b_\tau B_{xy} + b_{xy} B_\tau = \sum_{i=1}^{L_x} (W_{\tau,y;i+\frac{1}{2}}^b W_{y;i+\frac{1}{2}}^B + W_{\tau,y;i}^B W_{y;i}^b) + \sum_{j=1}^{L_y} (W_{\tau,x;j+\frac{1}{2}}^b W_{x;j+\frac{1}{2}}^B + W_{\tau,x;j}^B W_{x;j}^b) .$$

(A.14)

Following the above procedure, the partition function after gauging is

$$Z_{X/\mathbb{Z}_2^{\mathrm{sub}}}(W_{\tau,x;j}^B, W_{\tau,y;i}^B, W_{x;j+\frac{1}{2}}^B, W_{y;i+\frac{1}{2}}^B)$$
$$= \frac{1}{2^{2(L_x+L_y-1)}} \sum_{W_{\tau,x;j+\frac{1}{2}}^b, W_{\tau,y;i+\frac{1}{2}}^b, W_{x;j}^b, W_{y;i}^b = 0,1} Z_X(W_{\tau,x;j+\frac{1}{2}}^b, W_{\tau,y;i+\frac{1}{2}}^b, W_{x;j}^b, W_{y;i}^b)$$
$$\times (-1)^{\sum_{i=1}^{L_x}(W_{\tau,y;i+\frac{1}{2}}^b W_{y;i+\frac{1}{2}}^B + W_{\tau,y;i}^B W_{y;i}^b) + \sum_{j=1}^{L_y}(W_{\tau,x;j+\frac{1}{2}}^b W_{x;j+\frac{1}{2}}^B + W_{\tau,x;j}^B W_{x;j}^b)} .$$

(A.15)

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
