# Peer review of "Subsystem Non-Invertible Symmetry Operators and Defects"

_SciPost Physics_

## Round 2 · Referee Report · Anonymous (Referee 1) · 2023-6-14

Strengths

This paper makes a new generalization of the notion of symmetry.

Weaknesses

Relationships between different concepts are unclear.

Report

This paper presents a generalization of global symmetries, namely subsystem non-invertible symmetry. Especially, this paper studies the subsystem Kramers-Wannier duality in (2+1)-dimensional spin system and provides the detailed properties of duality operators (defects).

This paper points out a new important concept related to global symmetries. It provides excellent explanations and concrete examples. Therefore, I highly recommend publication of this paper in SciPost.

Requested changes

There are minor points that I would encourage the authors to address.

1 - There seems to be no definition of (H_i^t)^\dagger in the sentence above (2.21). Should it be replaced with (N_i^t)^\dagger?

2 - Even if the subsystem KW duality is a symmetry, subsystem KW duality operators (defects) are not recognized as subsystem symmetry operators (defects), opposed to their names. It is because the names of subsystem KW duality defects come from subsystem symmetry gauging, not from subsystem symmetry itself. If this understanding is correct, then I think that this point should be emphasized to avoid confusion.

3 - There are possibly grammatical errors or typos (pages and lines are according to the arXiv version.).
page 2, line 5: "more than one properties" -> "more than one property"
page 25, line 1: "partition function in" -> "partition function is"
page 29 and 30: The sentence "The number of holonomy variables grows with system size" is repeated. The first one seems unnecessary.

  • validity: high
  • significance: high
  • originality: high
  • clarity: good
  • formatting: excellent
  • grammar: excellent

Author:  Weiguang Cao  on 2023-08-08  [id 3880]

(in reply to Report 1 on 2023-06-14)

Dear referee,

We thank your effort and time on reading the draft. We fixed all typos you mentioned in comment 1 and 3.

Related to your second comment:
"2 - Even if the subsystem KW duality is a symmetry, subsystem KW duality operators (defects) are not recognized as subsystem symmetry operators (defects), opposed to their names. It is because the names of subsystem KW duality defects come from subsystem symmetry gauging, not from subsystem symmetry itself. If this understanding is correct, then I think that this point should be emphasized to avoid confusion."

Our response: we added a comment in the end of section 1.2
"Here, the subsystem KW duality symmetry has co-dimension 1 non-invertible symmetry operator and defect, which is different from the co-dimension 2 invertible subsystem $\mathbb Z_2$ symmetry operators and defects. Furthermore, the non-invertible fusion rule will mix operators (defects) of different co-dimensions."

We hope this will clarify the confusions.

Best,
Weiguang, Linhao, Masahito and Yunqin

---

## Round 2 · Referee Report · Anonymous (Referee 2) · 2023-6-23

Report

The paper discusses and constructs non-invertible subsystem symmetries in 2+1 dimensions. This is the first example of such a generalized symmetry. It is well-written and contains concrete examples. I recommend the draft for publication.

Requested changes

In both section 2 and 3, the authors discuss KW duality operator/defect 'N' and find its fusion rule in generality. However, they should emphasize that what they actually discuss is the KW interface 'N' between a theory before and after gauging. This interface has to be fused with an isomorphism 'U' which identifies the theory before and after gauging. The KW defect/operator that lives in a single theory is 'NU' rather than just 'N'.

For instance, the authors find the fusion rule

N x N^\dagger

which makes sense when 'N' is an interface. But they do not consider the fusion

NU x NU

which can be different from example to example. Authors should add a short discussion emphasizing where they discuss the interface and when discuss a defect/operator that exists in a single theory.

  • validity: -
  • significance: -
  • originality: -
  • clarity: -
  • formatting: -
  • grammar: -

Author:  Weiguang Cao  on 2023-08-08  [id 3881]

(in reply to Report 2 on 2023-06-23)

Dear referee,

We thank your effort and time on reading the draft. We include the following footnote in the introduction to clarify terminologies.

"We mainly use the terminology "operators" for mappings from one Hilbert space to another Hilbert space and "defects" for interfaces between two theories. However, in Sec.2.4 and 3.4, we also discussed about operators on one Hilbert space and defects within one theory."

We also include section 2.4 and 3.4 to discuss about operators and defects that exist in a single theory.

We hope this will clarify the confusions.

Best,
Weiguang, Linhao, Masahito and Yunqin

---

## Editorial Decision

resubmitted